# Endosome rupture enables enteroviruses from the family *Picornaviridae* to infect cells
Aygul Ishemgulova [1,3] ✉, Liya Mukhamedova[1,3], Zuzana Trebichalská[1,2,3], Veronika Rájecká [1], Pavel Payne[1], Lenka Šmerdová[1], Jana Moravcová[1], Dominik Hrebík[1], David Buchta[1], Karel Škubník[1], Tibor Füzik [1], Štěpánka Vaňáčová [1], Jiří Nováček [1] & Pavel Plevka [1] ✉

Membrane penetration by non-enveloped viruses is diverse and generally not well understood. Enteroviruses, one of the largest groups of non-enveloped viruses, cause diseases ranging from the common cold to life-threatening encephalitis. Enteroviruses enter cells by receptor-mediated endocytosis. However, how enterovirus particles or RNA genomes cross the endosome membrane into the cytoplasm remains unknown. Here we used cryo-electron tomography of infected cells to show that endosomes containing enteroviruses deform, rupture, and release the virus particles into the cytoplasm. Blocking endosome acidification with bafilomycin A1 reduced the number of particles that released their genomes, but did not prevent them from reaching the cytoplasm. Inhibiting post-endocytic membrane remodeling with wiskostatin promoted abortive enterovirus genome release in endosomes. The rupture of endosomes also occurs in control cells and after the endocytosis of very low-density lipoprotein. In summary, our results show that cellular membrane remodeling disrupts enterovirus-containing endosomes and thus releases the virus particles into the cytoplasm to initiate infection. Since the studied enteroviruses employ different receptors for cell entry but are delivered into the cytoplasm by cell-mediated endosome disruption, it is likely that most if not all enteroviruses, and probably numerous other viruses from the family *Picornaviridae*, can utilize endosome rupture to infect cells.

Enteroviruses from the family *Picornaviridae* cause a wide range of diseases, including the common cold, hand-foot-and-mouth disease, encephalitis, and paralysis, with major health, economic, and societal impacts[1–3]. Virions of enteroviruses are formed of capsids with diameters of 30 nm and ~7500 nucleotide-long single-stranded RNA genomes[4]. Enterovirus capsids are built from VP1, VP2, and VP3 subunits organized with pseudo-T = 3 icosahedral symmetry and minor capsid proteins VP4 symmetrically attached to the inner face of the capsid[4–6]. To initiate infection, enteroviruses bind to receptors and enter cells by endocytosis[7–13]. Endocytosis is an essential cellular process of membrane vesicle traffic, which involves the internalization of segments of the plasma membrane and engulfed extracellular cargo into a cell. There are multiple types of endocytosis that function in parallel and can be broadly divided into (i) macroscale phagocytosis or macro-pinocytosis and (ii) microscale clathrin-, caveolin-, and flotillin-mediated as well as coat-independent endocytosis[14]. Enteroviruses attach to specific cellular receptors; however, they can utilize various types of endocytosis for cell entry[7–9,15]. Rhinovirus A and B serotypes are classified into major and minor groups depending on whether they utilize intercellular adhesion molecule 1 or low-density lipoprotein receptor (LDLR) family members for cell entry[16–18]. Of the viruses analyzed in this study, rhinovirus 2 binds to LDLRs[17]. The entry of enterovirus 71 is mediated by P-selectin glycoprotein ligand 1, Scavenger receptor class B member 2, and heparan sulfate[19–21]. Echoviruses 18 and 30 attach to the neonatal Fc receptor, while echovirus 30 additionally utilizes a decay-accelerating factor[22,23]. In endosomes, enteroviruses are exposed to acidic pH[23–25]. In vitro studies have shown that receptor binding, acidic pH, and changes in ion concentrations induce the transformation of enterovirus virions into activated particles, which are characterized by expanded capsids, reduction of contacts between pentamers of capsid proteins, the release

[1]Central European Institute of Technology, Masaryk University, Kamenice 5, Brno, 625 00, Czech Republic. [2]National Centre for Biomolecular Research, Faculty of Science, Masaryk University, Kamenice 5, 625 00 Brno, Czech Republic. [3]These authors contributed equally: Aygul Ishemgulova, Liya Mukhamedova, Zuzana Trebichalská. ✉e-mail: aishemgulova@gmail.com; pavel.plevka@ceitec.muni.cz

**Fig. 1 | Rhinovirus 2 infection of a cell imaged using cryo-ET and cryo-EM. a** Tomographic reconstruction of the cytoplasm of an infected cos-7 cell. Positions of virions, empty particles, ribosomes (EMD-8345)[119], and microtubules (EMD-12639)[120] were identified using template matching as implemented in emClarity[121], and the corresponding high-resolution structures were placed into the tomogram. Microfilaments and membranes were segmented manually. The dashed rectangle indicates the position of the area shown at higher magnification in panel B. Scale bar 200 nm. **b** Detail of endosomes containing virions of rhinovirus 2. The central vesicle is ruptured at the place indicated with a black arrowhead. Scale bar 100 nm. **c, d** Surface representations of single-particle cryo-EM reconstructions of rhinovirus 2 virion (**c**) and empty particle (**d**) calculated from projection images of infected cos-7 cells. The surfaces of the particles are rainbow-colored according to their distance from the particle center. Bottom-right quadrants were replaced with central slices of cryo-EM density distributions within the particles to demonstrate that the virion contains a genome and the empty particle does not. Scale bar 10 nm.

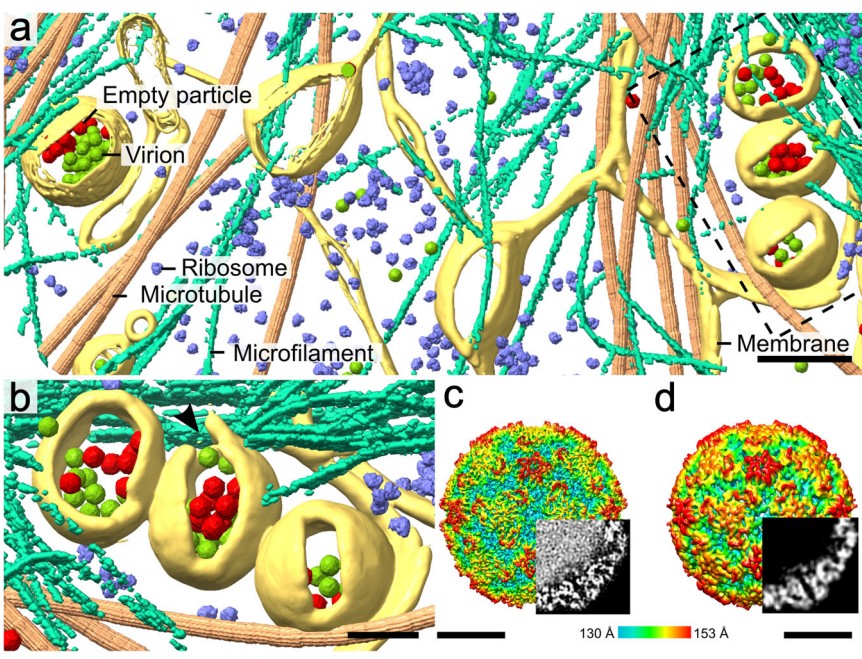

of VP4, externalization of N-termini of VP1, and granular distribution of genomic RNA[26–30]. Activated particles spontaneously release their genomes to enable the initiation of translation and replication[29–32]. The reactions of virions of rhinovirus 14 to the genome-inducing environmental stimuli were shown to be heterogeneous – some of the virions remained in native form, while others converted to activated particles or released their genomes[33].

Various mechanisms for enterovirus genome delivery into the cytoplasm have been proposed[8,34]. It was suggested that minor group rhinoviruses, including rhinovirus 2, release their genomes in endosomes and translocate them into the cytoplasm through transmembrane pores formed by VP4 and N-termini of VP1[35–38]. Poliovirus particles associate with liposomes decorated with extracellular domains of poliovirus receptors and form umbilical connections to the membrane through which the genomes might be transported[39–44]. However, diffusion of the ~7500-nucleotide-long RNA genome through an opening in a capsid and a putative transmembrane pore would be a slow process, which has not been verified in vivo. Major group rhinoviruses and rhinoviruses, which have adapted to use heparan-sulfate as a receptor were indicated to induce endosome disruption and release their genomes both into the endosomes and cytoplasm[45,46]. Despite the previous studies, our understanding of enterovirus genome delivery into the cell cytoplasm is incomplete.

Here we used cryo-electron tomography (cryo-ET) to visualize the cell entry of rhinovirus 2, echovirus 18, echovirus 30, and enterovirus 71 in situ. We show that enterovirus-containing endosomes deform, rupture, and release the virus particles into the cytoplasm. Blocking endosome acidification with bafilomycin A1 inhibited enterovirus genome release, but did not prevent endosome disruption. Inhibiting actin-dependent post-endocytic membrane remodeling with wiskostatin resulted in the accumulation of empty enterovirus particles in endosomes. We propose that cellular membrane-remodeling machinery disrupts endosomes and thus enables the diffusion of enterovirus particles and genomes into the cytoplasm.

## Results and Discussion
### Endocytosis of enteroviruses visualized in situ
We employed cryo-ET to visualize the entry of rhinovirus 2, echovirus 18, echovirus 30, and enterovirus 71 into fibroblast-like cos-7 cells (Figs. 1, 2, Supplementary Fig. 1–3, Supplementary Movie 1). The more than 250

recorded tomograms were of sufficient quality to enable the identification of genome-containing and empty enterovirus particles, ribosomes, microfilaments, microtubules, and cellular membranes (Fig. 1, Supplementary Movie 1). Seven to sixty minutes post-infection, enterovirus particles were both in membrane-bound vesicles and cytoplasm, indicating that enterovirus cell entry was a continuous and unsynchronized process. With prolonged time post-infection, the fractions of empty particles and particles in the cell cytoplasm increased (Supplementary Fig. 4). Some of the virus-containing endosomes were partially or fully covered with protein coats; however, we could not identify the type of vesicle coating (Fig. 2ab, Supplementary Fig. 1a, 2ab, 3a). Previous studies demonstrated that enteroviruses utilize clathrin-dependent, caveolin-dependent, and other types of endocytosis for cell entry[7–9,15].

### Enteroviruses detach from receptors in endosomes
The cell entry of enteroviruses is initiated by binding to specific receptors[4,17,19–23]. Accordingly, the studied enteroviruses attached to the receptors embedded in the cytoplasmatic membranes of cells (Supplementary Fig. S5). However, after endocytosis, all of the studied enteroviruses detached from receptors (Fig. 2a–f, Supplementary Figs. 1–3, 5, Supplementary Movie 2). Both genome-containing and empty particles were distributed randomly in the endosomes (Fig. 2a–f, Supplementary Fig. 1a–c, 2a–c, 3a–c). This confirms previous evidence that rhinovirus 2 detaches from LDLRs soon after endocytosis because of the conformational changes of the receptors induced by low calcium ion concentrations and acidic pH[47–51]. The mechanisms promoting the detachment of echovirus 18, echovirus 30, and enterovirus 71 from their respective receptors have not been identified. For these viruses, the dissociation is likely triggered by conformational changes to the receptors or virus capsid proteins induced by acidification or changes in ion concentrations in the lumen of maturing endosomes[33]. Our results provide evidence that detachment from receptors in endosomes is a common feature of enterovirus cell entry.

It has been shown previously that the N-termini of VP1 subunits, which are exposed at the surface of activated particles, can mediate membrane binding and could even induce membrane disruption[27,29,52–57]. Furthermore, VP4 subunits released from enterovirus activated particles were shown to form pores in membranes[58]. In vitro studies of poliovirus and

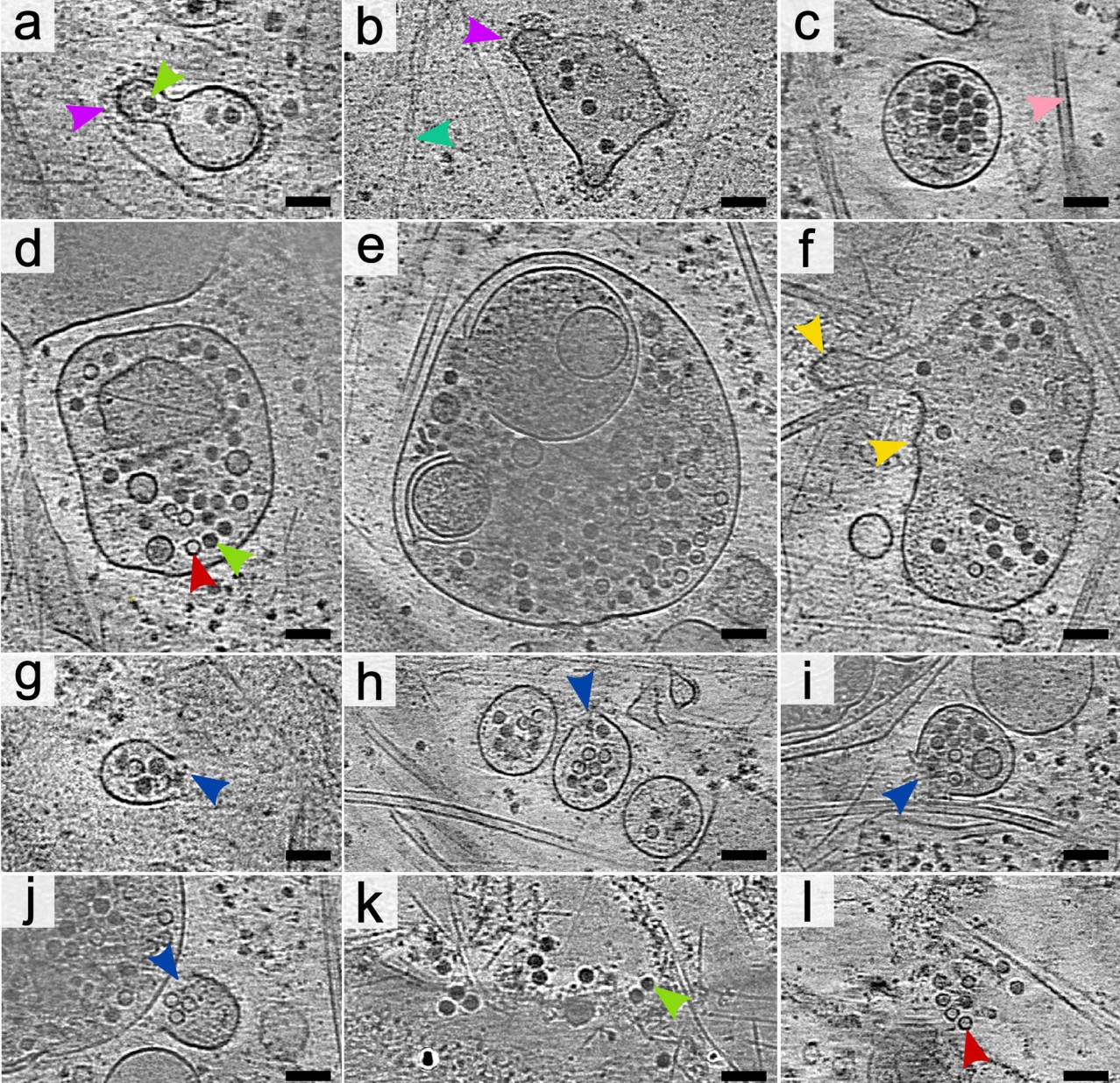

**Fig. 2 | Release of rhinovirus 2 from endosomes into the cytoplasm.** The images show 1.66 nm-thick tomographic slices of rhinovirus 2-infected cos-7 cells seven to sixty minutes post-infection. **a, b** Endosomes with segments of their surface covered with protein coats. Selected coated areas are indicated by magenta arrowheads. A selected virion is indicated by a green arrowhead. A selected actin microfilament is indicated by a cyan arrowhead. **c** Single-membrane endosome with a smooth surface containing virions and empty particles of rhinovirus 2. A pink arrowhead indicates a selected microtubule. **d, e** Multilamellar endosomes with a smooth outer membrane containing virions and empty particles. A selected empty particle is indicated by a red arrowhead. **f** Endosome with warped membrane forming cone-shaped protrusions. Selected tips of membrane warping are indicated by yellow arrowheads. **g–j** Ruptured endosomes release rhinovirus 2 into the cytoplasm. Positions of the openings in endosome membranes are indicated by blue arrows. **k, l** Virions and empty particles in the cell cytoplasm. Scale bar 100 nm.

HRV2 have shown the binding of activated particles to liposomes, and it was speculated that the transfer of RNA from the activated particles is mediated by umbilical connections up to 5 nm long[37,41,45,59]. However, cryo-tomograms of infected cells show nearly all enterovirus particles located more than 5 nm from the closest endosome membrane (Fig. 2, Supplementary Fig. 1–3, 6). With the exception of tight packing of particles in the minority of endosomes, enterovirus particles were distributed randomly in endosomes, and their mean distance from the closest membrane was 36 nm +/− 23 nm (Supplementary Fig. 6). These contrasting results indicate that enteroviruses may employ not only various receptors to attach to cells and different endocytic pathways to gain entry into host cells, but also multiple mechanisms to breach endosome membranes[15,57].

## Warping of endosome membranes

Some of the endosomes containing the studied enteroviruses formed cone-shaped protrusions, which we named membrane warping (Fig. 2f, Supplementary Fig. 1c, 2c, 3c, Supplementary Table 1). Membranes of a sub-population of endosomes in non-infected cells were also warped, indicating that this is a natural cellular process (Fig. 3a, Supplementary Fig. 7, Supplementary Table 1). The observed membrane warping may be related to the previously characterized formation of membrane protrusions, which enable early and sorting endosomes to sort and recycle cargo, receptors, and membranes[60–62]. The shapes of the membrane cones indicate that they are formed by an external force pulling the membrane away from the endosome center (Figs. 2f, 3a, Supplementary Fig. 1c, 2c, 3c, 7). In some cases, the tips of

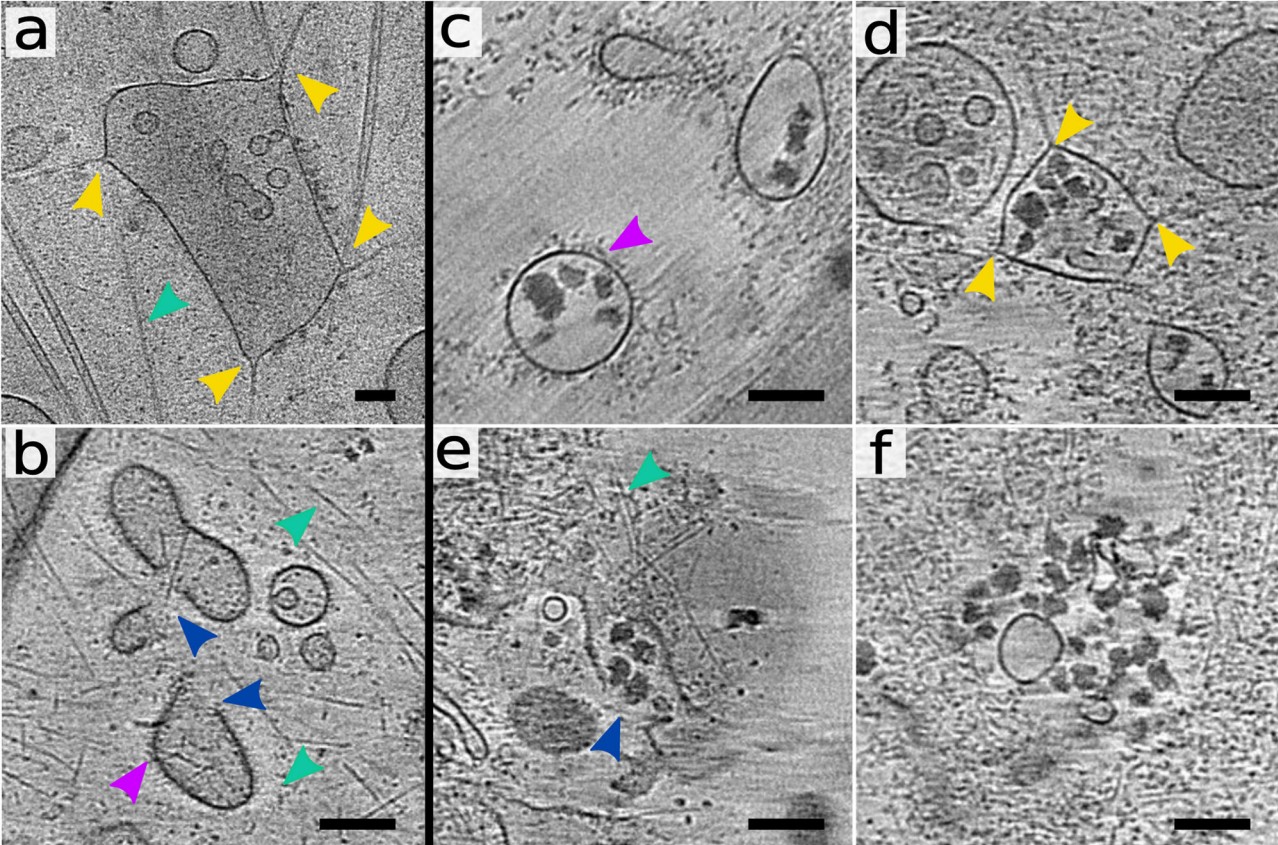

**Fig. 3 | Endosome rupture in uninfected cells and release of VLDL from ruptured endosomes.** The images show 1.66-nm-thick tomographic slices of cos-7 cells. **a, b** Warped (**a**) and ruptured (**b**) endosomes in control cells. The positions of membrane warping are indicated by yellow arrowheads, and the openings are indicated by blue arrowheads. Coated area of an endosome is indicated by a magenta arrowhead. Actin microfilaments are indicated by cyan arrowheads. **c–f** Sections from cryo-tomograms of cos-7 cells sixty minutes after the addition of OsO₄-stained VLDL complexes. **c** Single-membrane endosomes with smooth surfaces containing VLDL complexes. VLDL complexes are visible as electron-dense irregular objects. **d** Endosome with warped membrane forming cone-shaped protrusions. Selected tips of membrane warping are indicated by yellow arrowheads. Actin filaments attached to the tips of the membrane cones are visible. **e** Ruptured endosome releasing VLDL into the cytoplasm. The position of the opening in the endosome membrane is indicated by a blue arrowhead. **f** VLDL complexes in the cell cytoplasm. Scale bars 100 nm. Please note that panel **a** is shown at lower magnification than panels **b–f**.

the membrane cones were attached to actin filaments that may serve as scaffolds for putative complexes, which bind to and pull on the vesicle membranes (Supplementary Fig. 1c). It was shown that the remodeling of early and sorting endosomes is regulated by the interplay of numerous protein complexes, including endosomal sorting complexes required for transport (ESCTR)[63], retromer[64], retriever[65], nexins[66], neural Wiskott–Aldrich syndrome protein (N-WASP)[67,68], Arp 2/3[69], small GTP-binding proteins Rab[70], and actin[71]. It is likely that some of these membrane remodeling complexes contribute to the warping and disruption of enterovirus-containing vesicles, as is discussed below.

**Enterovirus particles and genomes are released into cytoplasm from ruptured endosomes**

The escape of enterovirus particles or genomes from endosomes is essential for the initiation of infection. Cells infected by rhinovirus 2, echovirus 18, echovirus 30, and enterovirus 71 contained ruptured endosomes with openings of sufficient size to enable diffusion of the virus particles or genomes into the cytoplasm (Figs. 1a, b, 2g–j, Supplementary Fig. 1d, 2d, 3d, Supplementary Movie 1, Supplementary Table 1). Numerous enterovirus particles were in the vicinity of the disrupted vesicles. The studied viruses bind to different receptors at the cell surface and belong to distinct clades within the genus *Enterovirus*. Nevertheless, our results provide evidence that they all employ the same mechanism of endosome escape. Therefore, it is likely that most, if not all enteroviruses can use endosome rupture to reach the cytoplasm.

Endosome disruption was previously proposed as a genome delivery mechanism for major group rhinoviruses and rhinoviruses adapted to use heparan-sulfate as a receptor[15,45,46]. In contrast, it was indicated that minor group rhinoviruses, including rhinovirus 2, release their genomes in late endosomes and translocate them into the cytoplasm through transmembrane pores[35,36]. These assumptions were based on in vitro analyses of the association of rhinovirus 2 with specific membrane fractions isolated from infected cells[35,36]. It is possible that the virus particles and cell membranes were affected by the isolation procedure and may not represent the rhinovirus 2 infection process in vivo.

Cryo-ET provides snapshots of cell and virus structures; however, it does not allow monitoring of the progress of infection of individual virus particles over time. Therefore, it cannot be determined which, if any, of the observed particles would successfully initiate infection. Moreover, the infectious unit-to-particle ratio of most enteroviruses is about 1 to 1,000[72–74]. To enable the visualization of enterovirus infection by cryo-ET, the cells were infected using a multiplicity of infection of 29 (please see Materials and methods for details). This high virus dose was required to enable the observation of multiple enterovirus particles in high-magnification tomograms, each of which captured only 0.17% of a cell. Therefore, our cryo-ET analyses do not provide proof that endosome disruption is the only enterovirus infection pathway. The high multiplicity of infection probably enabled enteroviruses to enter cells using multiple mechanisms of internalization, and the observed membrane rupture may not represent the most abundant physiological process in the context of a low multiplicity of

**Fig. 4 | Effects of inhibitors interfering with endosome maturation on enterovirus cell entry.** Panels **a–d** show 1.66-nm-thick tomographic slices from cos-7 cells pre-treated with the inhibitors for thirty minutes before infection and vitrified sixty minutes post rhinovirus 2-infection. **a, b** Cells treated with wiskostatin, an inhibitor of N-WASP-mediated actin polymerization that is required for post-endocytic membrane remodeling. **a** Oblique endosomes containing empty particles of rhinovirus 2. A red arrowhead indicates a selected empty particle. **b** A rare occurrence of ruptured endosome in wiskostatin-treated cos-7 cells. **c, d** Cells treated with bafilomycin A1, an inhibitor of endosome acidification. **c** Endosomes full of rhinovirus 2 virions. A green arrowhead indicates a selected virion. **d** Virions are released into the cytoplasm. Scale bar 100 nm. **e** Plot comparing differences in the probability of genome release from enterovirus particles in endosomes and cytoplasm of wiskostatin and bafilomycin A1-treated cells relative to the infection of untreated cells.

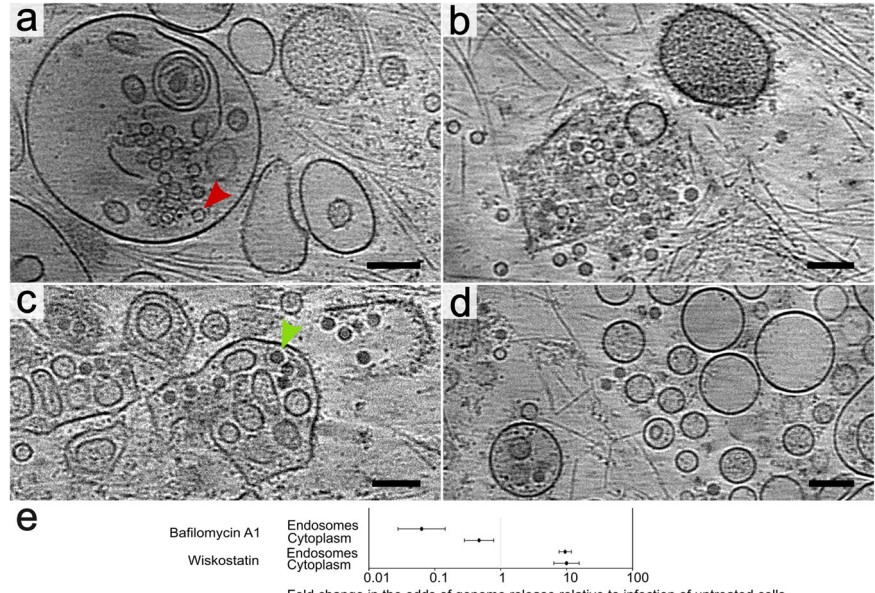

infection. Nevertheless, we demonstrate that enterovirus particles and genomes can be released from endosomes by membrane rupture, and that this mechanism is shared across the enterovirus genus. The involvement of membrane rupture in enterovirus infection under a low multiplicity of infection conditions is corroborated by the blockage of enterovirus infection by wiskostatin—an inhibitor of post-endocytic membrane remodeling, as is discussed below.

### Endosome rupture is triggered by a cellular mechanism

Lipid vesicles, including endosomes, are stable under physiological conditions, and their rupture requires a force that exceeds the forces holding the membrane together. There is evidence that the VP4 and N-termini of VP1 of some enteroviruses can permeabilize liposome and endosome membranes[34,35,38,42,75]. However, the membrane warping and the shapes of the ruptured endosomes in enterovirus-infected cells are evidence of an external force pulling on the endosome membrane – a mechanism incompatible with the action of VP4 and the N-termini of VP1 subunits. Furthermore, non-infected control cells contain a fraction of endosomes with ruptured membranes (Fig. 3b, Supplementary Fig. 7) and infected cells contain ruptured endosomes with no virus inside (Supplementary Fig. 8, Supplementary Table 1). This indicates that endosome disruption can be triggered by a cellular mechanism without the direct involvement of virus components. Enterovirus cell entry probably induces the concentration of virus-specific receptors in endosomes, which may stimulate signaling that activates cellular pathways that mediate endosomal rupture.

Rhinovirus 2 utilizes proteins from the family of LDLRs, including very low-density lipoprotein (VLDL) receptor, for cell binding and entry[17]. We investigated VLDL cell entry to enable the comparison of the effects of the virus and native cargo on endocytosis mediated by the same receptor. The various stages of the development of VLDL-containing endosomes are similar to those of endosomes that contain rhinovirus 2 (Fig. 3c–f). Inside the endosomes, VLDL complexes detached from receptors due to acidic pH and lowered calcium ion concentration[47–51] and were distributed randomly (Fig. 3c, d). Membranes of some VLDL-containing endosomes warped, and the tips of the membrane protrusions were attached to actin filaments (Fig. 3d). Finally, membranes of some VLDL-containing endosomes were ruptured (Fig. 3e), and VLDL diffused into the cytoplasm (Fig. 3f). Previous studies have shown that VLDL is transported to lysosomes for degradation and release of lipids and cholesterol to be incorporated into membranes[76]. Our results do not exclude this canonical pathway; however, we show that a fraction of VLDL is released into the cytoplasm by endosome rupture via a

mechanism morphologically similar to that utilized by enteroviruses. This provides evidence that endosomal rupture is a fundamental cellular process, which does not require enterovirus VP4 and VP1.

### Blockage of post-endocytic membrane remodeling inhibits enterovirus infection

To test the hypothesis that endosome disruption, mediated by a cellular pathway, enables enteroviruses to reach the cytoplasm, we blocked post-endocytic membrane remodeling with wiskostatin. The binding of wiskostatin to N-WASP stabilizes the protein in autoinhibited conformation, prevents it from activating the Arp2/3 complex, and thus blocks processes of cellular membrane remodeling that depend on actin polymerization[77,78]. However, wiskostatin treatment also causes a decrease in cellular ATP levels and inhibits other, N-WASP-independent functions[79]. Additionally to HRV2, echovirus 18 was selected for the wiskostatin inhibition studies because it propagated in cos7 cells most efficiently of the analyzed viruses. In wiskostatin-treated cells most of the particles were trapped inside oblique endosomes; however, some were released into the cytoplasm (Fig. 4). The infection of cells pre-treated with wiskostatin increased the odds of genome release inside endosomes 10-fold relative to untreated cells (Fig. 4e, Supplementary Table 2). The inhibition of endosomal membrane remodeling by wiskostatin probably resulted in a prolonged retention of enterovirus particles in endosomes, where they released genomes. Relative to one-hour post-infection, echovirus 18 replicated its genome 20- and 35-fold, seven and nine hours post-infection, respectively (Supplementary Fig. 9, 10, Supplementary Table 3). In contrast, in cos-7 cells pretreated with wiskostatin, the relative replication of echovirus 18 did not exceed 5-fold both seven and nine hours post-infection (Supplementary Fig. 9, 10). We speculate that the reduced efficiency of genome replication was caused by predominant genome release in endosomes, where the virus genomes may have been degraded by RNases. However, the efficiency of echovirus 18 genome replication may have been also reduced by decreased levels of proteosynthesis and RNA synthesis due to the low cellular ATP levels, which are a side effect of wiskostatin treatment[79]. Nevertheless, our results demonstrate that N-WASP-mediated actin polymerization is important for the disruption of endosomes, which enables the delivery of enteroviruses into the cytoplasm.

### Enteroviruses release genomes in endosomes and cytoplasm

Enterovirus genomes could be degraded by RNases when released from capsids, but it has been shown that the co-endocytosis of RNases with

infecting viruses does not reduce enterovirus infectivity[44,80]. However, the mechanism that protects enterovirus genomes from degradation remains unknown. Here we show that both endosomes and the cytoplasm of infected cos-7 cells contained full and empty particles of the studied enteroviruses (Figs. 1, 2, Supplementary Fig. 1–4, Supplementary Movie 1). In most cases, the cytoplasm contained a higher fraction of empty particles than endosomes (Supplementary Fig. 4), which provides evidence that enteroviruses release their genomes in the endosomes and cytoplasm, as indicated previously[15]. However, there is evidence that the productive genome release of poliovirus occurs across the membrane of vesicles or tightly sealed membrane invaginations[57]. It is possible that enteroviruses employ multiple pathways for genome delivery, as discussed below.

The variability in the location and timing of genome release among individual enterovirus particles is probably caused by the variability in environmental stimuli experienced by individual particles and by internal differences between the particles. Individual particles bind to distinct numbers of receptors at the plasma membrane, and therefore receive distinct levels of stimulation[33,81]. Furthermore, particles in different endosomes are exposed to distinct pH levels and ionic compositions, depending on the maturation state of the particular endosome[82]. The duration of their exposure to those endosome conditions also differs. Enterovirus virions may differ in the arrangement of packaged genomes and contain variable amounts of polyamines and inorganic cations that neutralize the repulsive interactions of virus RNA[83]. Finally, the timing of genome release from a particular particle is subject to capsid breathing and genome rearrangements caused by thermal motions. It is possible that at least some of the genome-containing particles observed in our cryo-EM experiments in the cytoplasm would not be able to release their genomes and initiate infection. The inability of some of the particles to release their genomes may contribute to the high ratio of particles to infectious units of enteroviruses[33,72,73].

The studied enteroviruses differed in the speed of accumulation of empty particles in infected cells. The fastest to release its genome was echovirus 18, with 84% empty particles in the cytoplasm thirty minutes post-infection, and the slowest was enterovirus 71, with 2% empty particles sixty minutes post-infection (Supplementary Fig. 4). The differences may be caused by (i) the viruses utilizing distinct receptors for cell entry, which may affect the speed of endocytosis and conditions in endosomes[84], (ii) different particle stability, which affects the timing of the initiation of genome release[85], and (iii) differences in the stability of empty particles of the enteroviruses, which may fall apart and thus would not be included in our statistics.

Some enterovirus-containing endosomes ruptured (Fig. 1, Supplementary Fig. 1–3, Supplementary Movie 1), releasing virions and free genomes into the cytoplasm, which may cause infection. The cell compartment where a genome is released probably influences the efficiency of infection, since the endosome lumen contains RNases that may cleave enterovirus genomes[80]. The release of the genome in the cytoplasm may have a higher probability of successful initiation of infection, because it reduces the chances of genome degradation due to the dilution of the RNases from endosomes in the cytoplasm.

The possibility of the initiation of the infection by genome release in the cytoplasm raises the question of how newly assembled enterovirus virions are protected from premature uncoating. We speculate that the newly assembled virions do not release their genomes because they have not been stimulated by environmental factors, including receptor binding, acidic pH, and an ion composition distinct from that of the cytoplasm[26–30].

Compounds binding to hydrophobic pockets of VP1 block the activation of enterovirus particles and genome release, and are efficient inhibitors of enterovirus infection[86–93]. Some of these inhibitors alter the conformation of the base of the receptor binding "canyon," forming the roof of the pocket, and block receptor binding[94]. Other capsid binding inhibitors allow receptor binding but prevent the conversion of virions to activated particles[95–97]. Virions that cannot release their genomes are probably degraded by the cells that endocytosed them.

## Activated particles are short-lived precursors of genome release in vivo

Cryo-tomograms of infected cells enable the straightforward identification of enterovirus particles and the determination of whether they contain genomes (Fig. 2, Supplementary Fig. 1–3). However, distinguishing enterovirus virions from activated particles, both of which contain genomes, requires high-resolution information, since the two structures differ only in the details of their capsids (Supplementary Table 4)[26,30,98]. Single-particle reconstruction of rhinovirus 2 using electron micrographs collected on lamellipodia of infected cos-7 cells enabled the determination of the structures of virions and empty particles to resolutions of 4.7 and 7.1 Å, respectively (Fig. 1cd, Supplementary Fig. 11, Supplementary Table 5). The classification procedures employed during the reconstruction of the genome-containing particles did not identify a sub-class of activated particles. Therefore, the infected cells probably only contained a small fraction of activated particles, which indicates that they are short-lived and rapidly release their genomes to become empty capsids. The fast release of genomes from activated particles in vivo contrasts with numerous previous in vitro studies in which the activated particles were sufficiently stable to enable the determination of their structures by cryo-EM or X-ray crystallography[28,30,31,33]. However, these experiments were performed at non-physiological pH, ion concentrations, and temperatures that may have stabilized the activated particles. Rapid genome release may be beneficial for enteroviruses in vivo because it accelerates the onset of virus translation and replication.

## Endosome acidification is not required for enterovirus escape from endosomes

Compounds that prevent the acidification of endosomes, including bafilomycin A1, block enterovirus infection to various extents[24,99–101]. In vitro experiments have demonstrated that acidic pH induces the formation of activated particles and genome release of many enteroviruses, which is likely prevented by bafilomycin A1[26–32]. Here we show that the treatment of cells with bafilomycin A1 leads to a 15-fold reduction in the odds of rhinovirus 2 genome release in the endosomes (Fig. 4ce). However, particles of rhinovirus 2 were released into the cytoplasm in bafilomycin A1-treated cells (Fig. 4d), and the odds of their genome release into the cytoplasm were only reduced 2-fold relative to the untreated cells (Fig. 4de). The observations that bafilomycin A1 treatment (i) does not prevent rhinovirus 2 escape from endosomes and (ii) has only a moderate effect on the probability of genome release in the cytoplasm, explain the limited effect of the compound on enterovirus infection described previously[24,99–101]. Furthermore, our results provide evidence that enterovirus escape from endosomes is independent of the activation of enterovirus particles and genome release.

## Summary

Based on previous results and our current analysis of enterovirus cell entry, we propose the following mechanism of enterovirus genome delivery: Enteroviruses enter cells by receptor-mediated endocytosis (Fig. 5)[7–9,15]. In endosomes, enteroviruses detach from receptors, and some of the particles release their genomes. However, the released genomes may be degraded by co-endocytosed RNases[80]. N-WASP-mediated actin polymerization[67,68] contributes to the warping and rupture of enterovirus-containing endosomes, demonstrating that enteroviruses can utilize a cellular mechanism for gaining entry into the cytoplasm (Fig. 5). Enterovirus virions and free genomes diffuse from the ruptured endosomes to the cytoplasm, where additional virions release their genomes. The cytoplasmic genome release may have a higher probability of resulting in successful infection than the endosomal one, because the RNases released from endosomes disperse in the cell cytoplasm. Since the studied enteroviruses employ distinct receptors for cell entry but are all delivered into the cytoplasm by cell-mediated endosome disruption, we propose that most if not all enteroviruses, and probably numerous other viruses from the family *Picornaviridae*, can use endosome rupture to reach the cytoplasm.

**Fig. 5 | Overview of enterovirus cell entry.** (1) Enterovirus virions bind to receptors at the cell surface and are endocytosed. (2) In endosomes, enteroviruses detach from receptors and are exposed to acidic pH. Some of the particles release their genomes. (3) Warping of membranes of virus-containing endosomes. (4) Rupture of endosome membrane, which depends on N-WASP-regulated actin polymerization, enables the diffusion of virions into the cytoplasm where additional particles release their genomes.

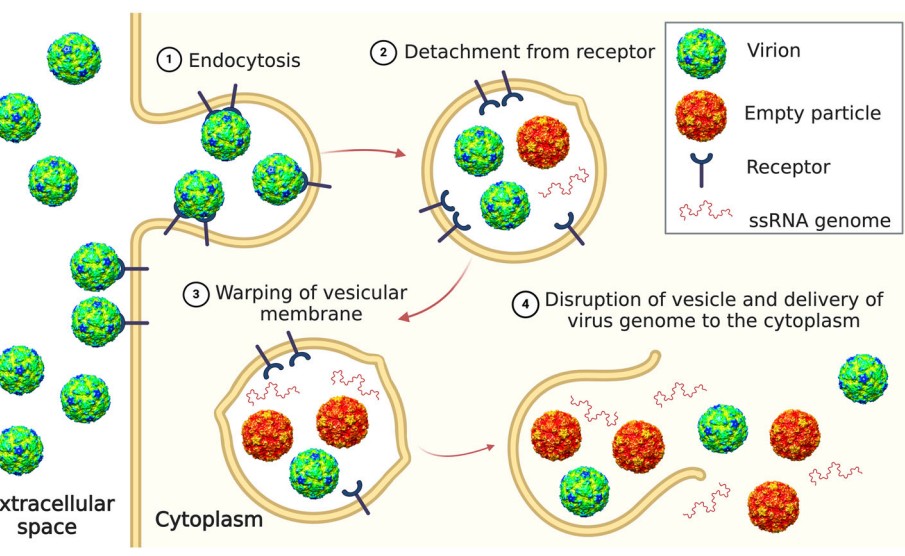

## Methods

### Production and purification of enteroviruses

Rhinovirus 2 (strain HGP, ATCC-482) was propagated in HeLa (ATCC-CCL2) cells; echovirus 18 (strain METCALF, ATCC-VR-852), echovirus 30 (strain Bastianni, ATCC-VR-1660), and enterovirus 71 (strain MY104-9-SAR-97[102]) were propagated in GMK cells (ATTC-CCL-81) cultivated in Dulbecco's modified Eagle's medium enriched with 10% fetal bovine serum (Sigma, F7534). For virus preparation, 50 tissue culture dishes with a diameter of 150 mm of cells grown to 100% confluence were infected with virus inoculum with a multiplicity of infection of 0.01 to 0.1 in DMEM with 3% fetal bovine serum. The infection was allowed to proceed for 24–72 h, at which point more than 90% of the cells exhibited the cytopathic effect. Any remaining attached cells were removed from the dishes using cell scrapers, and the cell media were harvested. The cell suspension was centrifuged at $15,000 \times g$ in a Beckman Coulter Allegra 25 R centrifuge, TA-10-250 rotor at $10\,°C$ for 30 min. The resulting pellet was resuspended in 10 mL of phosphate-buffered saline (PBS) ($10\,mM\ Na_2HPO_4$, $1.8\,mM\ KH_2PO_4$, $137\,mM\ NaCl$, and $2.7\,mM\ KCl$, pH 7.4, Sigma-Aldrich). The solution was subjected to three rounds of freeze-thawing by transfer between $-80$ and $37\,°C$, and homogenized using a Dounce tissue grinder. Cell debris was separated from the supernatant by centrifugation at $4300 \times g$ in a Beckman Coulter Allegra 25 R centrifuge, TS-5.1-500 rotor at $10\,°C$ for 10 min. The resulting supernatant was combined with media from the infected cells. Virus particles were precipitated by the addition of PEG-8000 and NaCl to final concentrations of 5% (w/v) and 0.5 M, respectively. The precipitation was allowed to proceed at $10\,°C$ with mild shaking (80 RPM) for 12 h. The following day, the precipitate was pelleted at $15,000 \times g$ in a Beckman Coulter Allegra 25 R centrifuge, TA-10 rotor at $10\,°C$ for 30 min. The supernatant was discarded, and the pelleted white precipitate was resuspended in 12 mL of PBS. $MgCl_2$ was added to a final concentration of 5 mM, and the sample was subjected to DNAse ($10\,\mu g/mL$ final concentration) and RNAse ($10\,\mu g/mL$ final concentration) treatment for 30 min at ambient temperature. Subsequently, trypsin was added to a final concentration of $0.5\,\mu g/mL$, and the mixture was incubated at $37\,°C$ for 10 min. pH 9.5 EDTA was added to a final concentration of 15 mM, and non-ionic detergent, NP-40™ (Sigma Aldrich Inc.), was added to a final concentration of 1%. The mixture was spun down at $4300 \times g$ in a Beckman Coulter Allegra 25 R centrifuge, TS-5.1-500 rotor at $10\,°C$ for 10 min. Particles from the resulting supernatant were pelleted through a 30% (w/v) sucrose cushion in resuspension buffer (0.25 M HEPES, pH 7.5 and 0.25 M NaCl) by centrifugation at $280,000 \times g$ in an Optima X 80 ultracentrifuge using a Beckman Coulter™ type 50.2 Ti rotor at $10\,°C$ for 2 h. The pellet was resuspended in 1.5 mL of PBS and loaded onto 60% (w/w) CsCl solution in PBS. The CsCl gradient was established by ultracentrifugation at $220,000 \times g$ in an Optima X80 ultra-centrifuge using a Beckman Coulter™ SW41 Ti rotor at $10\,°C$ for 18 h. The opaque bands containing the virus were extracted with a 20-gauge needle mounted on a 5 mL disposable syringe. The virus was transferred into PBS by multiple rounds of buffer exchange using a centrifugal filter device with a 100-kDa molecular weight cutoff.

### Preparation of cryo-EM samples of cos-7 cells infected by enteroviruses

The 200 mesh holey carbon grids for transmission cryo-EM (Quantifoil, Au, R2/1) were sterilized with UV for 20 min and incubated in fetal bovine serum (Sigma) for 15 min to increase cell adhesion. The grids were placed into Thermo Scientific™ Nunc™ Lab-Tek™ Chamber Slide™ cell culture dishes, and 80 μL of cos-7 cell suspension ($10^5$ cells / ml in Dulbecco's modified Eagle's medium enriched with 10% fetal bovine serum) were added to each well and cultivated for 16 h. Cells were washed twice with PBS and infected with about $5*10^{10}$ enterovirus particles in 5 μL of PBS (rhinovirus 2, echovirus 18, echovirus 30) or DMEM (enterovirus 71) per well. The infection was carried out at $37\,°C$ and 5% $CO_2$ for 5 min. Subsequently, the virus suspension was removed, fresh DMEM was added, and the cells were incubated at $37\,°C$ and 5% $CO_2$ for the desired time. Not all virus particles from the inoculum attached and entered into the cells. Identification of enterovirus particles in tomograms of infected cells enabled the calculation of the multiplicity of infection. The area of an average cos-7 cell was $2,500\ \mu m^2$ (Supplementary Fig. 12) and the area of a reconstructed tomogram was $4,36\ \mu m^2$. On average there were fifty enterovirus particles in a tomogram of an infected cell (Supplementary Table 2, line infected cells un-treated by inhibitors). Therefore, an average cell was infected by 29,000 virus particles. For enteroviruses, the ratio of infectious units to particles is approximately $1 : 1000$[72–74]. Therefore, the cells in our cryo-ET experiments were infected with a multiplicity of infection of 29. The high virus dose was required to enable observation of a representative number of enterovirus particles in the high-magnification tomograms, each of which represents only 0.17% of the area of an average cos-7 cell.

At the desired time post-infection, the cells were washed three times with PBS preheated to $37\,°C$. PBS was removed by pipetting, and the grids were vitrified using a Vitrobot Mark IV. Cells were vitrified at 7-, 20-, 30- and 60-min post-infection. Control cells were vitrified using the same procedure, except for adding the virus.

### Preparation of cryo-EM samples of cos-7 cells with VLDL

To enable the identification of very low-density lipoproteins (VLDL) in tomograms of cells, VLDL complexes were stained with osmium tetroxide

using the following procedure: Twenty microliters of PBS containing VLDL at a concentration of 1.1 mg/ml (Calbiochem) were placed as a hanging drop into a 3.4 ml sealed container with 100 µl of 4% osmium tetroxide solution. The staining, using the vapor diffusion of osmium, was allowed to proceed for 8 min, during which the VLDL solution turned brown. Five microliters of osmium tetroxide-stained VLDL solution were added to cos-7 cells grown on 200 mesh holey carbon grids, as described above for the preparation of enterovirus-infected cells. The cells were incubated at 37 °C for 5 min. Subsequently, the cells were washed three times with PBS preheated to 37 °C and incubated in DMEM at 37° and 5% $CO_2$ for 55 min. After the incubation, the cells were washed three times with PBS and vitrified, as described above.

### Preparation of cryo-EM samples of cos-7 cells treated with wiskostatin or bafilomycin A1 and infected by rhinovirus 2

The protocols used to study the effects of wiskostatin and bafilomycin A1 differed only in the concentration of the inhibitors, which were 50 µM for wiskostatin and 200 nM for bafilomycin. Cos-7 cells were grown on 200 mesh holey carbon grids, as described above for the preparation of enterovirus-infected cells. The cells were washed three times with PBS and then incubated in 5 µl of wiskostatin or bafilomycin dissolved in DMEM for 30 min. Subsequently, the medium covering the cells was replaced with 5 µl of DMEM containing rhinovirus 2 ($5*10^{10}$ enterovirus particles) and wiskostatin or bafilomycin. After 5 min of incubation, the cells were washed with PBS and incubated in DMEM for another 55 min. After the incubation, the cells were washed three times with PBS and vitrified, as described above.

### Calculating the effect of wiskostatin and bafilomycin A1 on the probability of enterovirus genome release

To determine the effect of wiskostatin and bafilomycin A1 on the probability that enterovirus particles release their genome, we calculated odds ratios (ORs) of full-to-empty particle conversion in drug-treated samples relative to an untreated control. Each odds ratio was calculated separately for each drug treatment and for particles in cytoplasm and in vesicles. For each odds ratio we performed statistical analysis and calculated 95% confidence intervals (95% CI), standard error (SE), standard normal deviate (z) and $p$ value (P). In wiskostatin-treated cells both in cytoplasm and in vesicles, the odds ratios showed a 10-fold increase in the probability of genome release relative to the control (cytoplasm: OR = 9.93, 95% CI 6.38–15.46, SE = 0.23, z = 10.17, $P < 0.0001$; vesicles: OR = 9.48, 95% CI 7.65–11.74, SE = 0.11, z = 20.60, $P < 0.0001$). In bafilomycin A1-treated samples, the odds ratio of genome release was reduced approximately two-fold in cytoplasm (OR = 0.47, 95% CI 0.28–0.78, SE = 0.26, z = 2.90, $P = 0.0037$) and about 15-fold in vesicles (OR = 0.063, 95% CI 0.028–0.144, z = 6.57, $P < 0.0001$). Particle counts and the numbers of tomograms analyzed for each treatment are listed in Supplementary Table 2.

### Determining the effect of wiskostatin on enterovirus infection using reverse-transcription quantitative PCR (RT-qPCR)

Cos-7 cells were grown in 9.5 $cm^2$ dishes to 80–90% confluency. Thirty minutes before infection, wiskostatin was added to the cell media to a final concentration of 50 µM. Subsequently, the cells were infected with echovirus 18 with a multiplicity of infection of 1–2. The cells were harvested 1, 7, and 9 h post-infection. To isolate RNA, cells from one plate were lysed using 1 ml of TriPure reagent. The total RNA was isolated according to the manufacturer's instructions (Roche). Total RNA was treated with 4 units of TurboDNase (AppliChem) followed by phenol : chloroform : isoamyl alcohol extraction and ethanol precipitation. The precipitated RNA was dissolved in RNAse-free water (Thermo Scientific). 1.5 µg of total RNA was used for cDNA preparation by reverse transcription (RT) with random hexamer primers and SuperScript III Reverse Transcriptase according to the manufacturer's protocol (Invitrogen). The RT-qPCR was performed using a Roche LC 480 (Roche) RT-qPCR machine in FastStart SYBR Green Master 480 (Roche) with gene-specific primer pairs complementary to two different positions in echovirus 18 genome named V3 and V5, *glyceraldehyde-3-phosphate dehydrogenase* (GAPDH), and *β-2 microglobulin* (B2M) (Supplementary Table 3). Relative transcript abundance was calculated by the ΔΔCt method[103]. Data are displayed as mean values normalized to the housekeeping gene (GAPDH, B2M) and time point of 1 h post-infection. The experiment was performed in three independent biological replicates. Results are shown as means and standard errors of the mean. p-values were calculated using a two-tailed Student's *t*-test, and the error bars show the standard deviation with a $p$ value ***$p < 0,005$.

### Statistics and Reproducibility

The effect of wiskostatin on enterovirus infection was analyzed using RT-qPCR and statistical analysis. The experiment was performed in three independent biological replicates. The raw data dataset and the complete analysis of the RT-qPCR data, using the standard Livak method, is provided in supplemental information. The data are first normalized to the housekeeping gene (GAPDH or B2M), which provides the dCt value. A second normalization to t = 1 h provides the ddCt value, which is then recalculated into fold change representing relative enrichment. Therefore, the data point t = 1 h has a relative value 1.

### Acquisition and analysis of cryo-electron tomography data

Tomographic tilt series of infected cells were collected using a ThermoFisher Titan Krios transmission electron microscope operating at 300 kV. The data were collected using a post-GIF Gatan K2 (Bioquantum 967) direct electron detector operating in zero-loss imaging mode with the width of the energy-selecting slit set to 20 e⁻V. The grid was first screened to localize cells with thin lamellipodia suitable for cryo-ET data collection (Supplementary Fig. 12). The tilt series were collected using SerialEM software[104] with a dose-symmetric tilt scheme[105] with an angular range of ±52° (2° increment) or ±60° (3° increment). With both settings, the total electron exposure was 50 e⁻/Å². Individual images were saved as movies of 4 frames in counting mode. The data were collected at a magnification of 26,000x, corresponding to a pixel size of 5.534 Å, and with defocus values ranging from −2.5 to −5.5 µm. Frames were aligned to compensate for drift and beam-induced motion during image acquisition using the program MotionCor2[106]. The resulting tilt series were aligned in IMOD using the patch tracking or fiducial marker-based alignment algorithms, binned three times, and the tomograms were calculated by weighted back-projection with a SIRT-like filter as implemented in the software package IMOD[107]. Tomograms were denoised using the Topaz unet-3d-10a model[108].

### Identification of the positions of virions, empty particles, ribosomes, and microtubules in tomograms

Aligned tilt series from IMOD were imported into an emClarity work environment, and the contrast transfer function (CTF) of the tilts was determined using emClarity (v 1.5.3.11). CTF-corrected tilt series were binned three times and reconstructed into tomograms (pixel size 16.57 Å). Template matching was performed by emClarity using resampled maps of ribosomes (EMD-8345), microtubules (EMD-12639), or HRV2 particles as templates. The set of template-matched particles was curated by inspecting each matched particle's position in the tomogram. False-positive matches were removed. Positions and orientations of the matched particles were exported from emClarity into a homemade program (available at https://github.com/fuzikt/tomostarpy; placeback_subvolume.py), which was used to position the maps of the search templates into a volume that matched the size of the original tomogram. This resulted in separate maps of positioned maps of ribosomes, microtubules, and rhinovirus 2 virions and empty particles. The maps of the positioned components were overlaid in ChimeraX[109] together with volumes of manually segmented membranes, actin filaments, and the original tomogram.

### Segmentation of cellular membranes and actin microfilaments in tomograms

The segmentation was performed using the software Amira (Thermo Fisher Scientific, version 2020.2). The membranes and filaments were manually

segmented using the Brush tool and masking to select voxels belonging to membranes or microfilaments. The segmented labels representing membranes and microfilaments were then expanded using the feature "Grow selection in volume". The labels were exported as three-dimensional maps in MRC format. Subsequently, the separate maps of membranes and microfilaments were overlaid with the volumes of virions, empty particles, ribosomes, and microtubules and the original tomogram using ChimeraX (UCSF ChimeraX, version 1.3 (2021-12-08))[109]. Pictures and movies were prepared using ChimeraX and the program FFMPEG.

### Acquisition and analysis of single-particle data

The data for single-particle analysis were collected using the same microscope and camera as with the cryo-ET data. Electron micrographs were recorded in the counting mode of the detector at a magnification of 105,000×, which corresponds to a pixel size of 1.34 Å. Total electron exposure was 50 e⁻ / Å² , and defocus values ranged from −1.8 to −3.0 μm. Movies of 40 frames were recorded for each 10 s of exposure. The frames from each exposure were aligned to compensate for drift and beam-induced motion using the program MotionCor2[106]. The resulting dose-weighted sums of aligned frames were used in the subsequent image processing steps, except for estimating contrast transfer function (CTF) parameters, which were determined from non-dose-weighted micrographs using the program gCTF v1.06[110,111].

Images of 2035 particles of rhinovirus 2 were manually boxed from micrographs of cells 30 and 40 min post-infection using cryolo_boxmanager from the software Cryolo 1.5.4[112]. The cellular localization of all of the boxed particles could not be determined with certainty due to limited fields of view and low contrast in the electron micrographs. Particles from within vesicles and cytoplasm were combined for the reconstruction. Particles were extracted with a box size of 448 pixels using RELION[113]. 2D classification was employed to separate 1590 full and 332 empty particles. Initial models of full and empty particles with imposed icosahedral symmetry were independently generated de novo using Relion's 3D initial model procedure. Subsequently, particles were subjected to several rounds of 3D classification using the initial models as references. Images of empty particles were binned to 224 pixels. Images of 1424 full particles and 332 empty particles were selected for reconstruction using the Relion 3D autorefine procedure. The resulting maps were used as initial models for subsequent 3D refinements. Icosahedral symmetry was imposed on the volumes during the refinement process. The final volumes were subjected to post-processing in RELION. The resolutions of the reconstructions were estimated according to the FSC 0.143 criterion.

To identify the putative sub-population of activated particles, all genome-containing particles were subjected to 3D classification using maps generated from PDB structures of the rhinovirus 2 virion (PDB: 1FPN) and activated particle (PDB: 4L3B) as starting models. Particles from the resulting classes were refined separately. In all cases, the reconstructed maps corresponded to virions, or the resolution of the reconstructions was insufficient to distinguish whether the structure belonged to a virion or activated particle.

### Comparison of maps for identification of particle type

The cryo-EM reconstruction of an empty particle of rhinovirus 2 was used for calibrating pixel size by comparing it to a map calculated from the PDB structure of an empty particle (PDB: 3TN9). The PDB structure was converted to a map using the software package XMIPP[114]. The pixel size of the cryo-EM map of the empty particle was changed in small steps, and each time a cross-correlation coefficient comparing it to the map calculated from the PBD structure was calculated using the program Mapman[115]. The highest cross-correlation was obtained for a pixel size of 1.324 Å. Previously determined PDB structures of the rhinovirus 2 virion (PDB: 1FPN)[98], activated particle (PDB: 4L3B)[30], and empty particle (PDB: 3TN9)[26] were converted to maps. The capsid cavity of cryo-EM reconstructions of the genome-containing particle was masked by applying a spherical mask with a diameter of 196 Å using the programs RELION 3.0.8[113] and EMAN2[116]. The

cryo-EM reconstructions and PDB-generated maps were low-pass filtered to a resolution of 4.7 Å using the program UCSF Chimera[117], and normalized using the program Mapman[115]. Correlation coefficients comparing cryo-EM reconstructions and maps generated from PDB structures were calculated using the program Mapman[115].

### Model building

The PBD structures of the virion (PDB: 1FPN)[98] and empty particle (PDB: 3TN9)[26] of rhinovirus 2 were rigid-body fitted into cryo-EM maps using the program Phenix[118]. The structures were subjected to coordinate and B-factor refinement using the program Phenix[118].

### Reporting summary

Further information on research design is available in the Nature Portfolio Reporting Summary linked to this article.

### Data availability

Cryo-EM electron density maps have been deposited in the Electron Microscopy Data Bank, https://www.ebi.ac.uk/pdbe/emdb/ under accession numbers EMD-15710 (rhinovirus 2 virion) and EMD-15711 (empty particle), and the fitted coordinates have been deposited in the Protein Data Bank, www.pdb.org under PDB ID codes 8AY4 and 8AY5, respectively. Tilt series and the corresponding reconstructed tomograms we deposited in the EMPIAR database using the following accession codes: rhinovirus 2 infected cells EMPIAR-12266 and EMPIAR-12270; rhinovirus 2 infected cells treated with bafilomycin A EMPIAR-12267; rhinovirus 2 infected cells treated with wiskostatin EMPIAR-12268; echovirus 18 infected cells EMPIAR-12260 and EMPIAR-12263; enterovirus 71 infected cells EMPIAR-12261 and EMPIAR-12271; echovirus 30 infected cells EMPIAR-12269; VLDL-treated cells EMPIAR-12262; and control cell EMPIAR-12258. The authors declare that all other data supporting the findings of this study are available within the article and Supplementary Data 1 file (source data for plots and statistics presented in the paper) or are available from the authors upon request.

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

## Acknowledgements
We gratefully acknowledge the Cryo-electron Microscopy and Tomography, Biomolecular Interactions and Crystallization, and Proteomics core facility of CEITEC supported by MEYS CR (LM2018127). We wish to thank Alex J. Noble for helpful discussions. This research was carried out under the project CEITEC 2020 (LQ1601), with financial support from the MEYS of the Czech Republic under National Sustainability Program II. Computational resources were supplied by the project "e-Infrastruktura CZ" (e-INFRA LM2018140) provided within the program Projects of Large Research, Development and Innovations Infrastructures. The research leading to these results received funding from the Czech Science Foundation, grant GX19-25982X to PP, 23-07372S to Š.V., and the National Institute of Virology and Bacteriology (Program EXCELES, ID Project No. LX22NPO5103) Funded by the European Union - Next Generation EU. The project New Technologies for Translational Research in Pharmaceutical Sciences /NETPHARM, project ID OP JAC CZ.02.01.01/00/22_008/0004607, is co-funded by the European Union. The project RNA for therapy, project ID OP JAC CZ.02.01.01/00/22_008/0004575, is co-funded by the European Union.

## Author contributions
A.I. and P.Pl. designed research; A.I., L.M., Z.T., V.R., L.Š., J.M., K.Š., and J.N. performed research; A.I., L.M., Z.T., V.R., P.Pa., D.H., L.M., D.B., T.F., Š.V., J.N., and P.Pl. analyzed data; and A.I., L.M., Z.T., V.R., P.Pa., D.H., T.F., Š.V., and P.Pl. wrote the paper.

## Competing interests
The authors declare no competing interests.
