## [Transparent Peer Review file · Communications Biology]

Endosome rupture enables enteroviruses to infect cells

Corresponding Author: Dr Pavel Plevka

Version 0:

Reviewer comments:

Reviewer #1

(Remarks to the Author)

Reviewer response:

The manuscript by Ishemgulova et al., focuses on imaging entry by enteroviruses into cells. Direct data on how non-enveloped viruses, such as enteroviruses, deliver their genome into the cytoplasm in cells is sparsely available. In this work, the authors have used cryo-electron tomography and imaged thin, cellular extensions at the periphery of cos-7 cells to observe multiple endocytosed enteroviruses and their release into the cytoplasm. The authors have used rhinovirus 2, echovirus 18, echovirus 30, and enterovirus 71 viruses in this study and imaged the infected cell peripheries at increasing time points post-infection. Based on their results, the authors put forth a general mechanism for enterovirus entry into cellular cytoplasm and discuss that these viruses exploit a cellular mechanism for membrane remodeling that does not directly involve any of the viral proteins for endosomal release. Enterovirus particles and genome escape into the cytoplasm by endosome rupture, irrespective of which receptor they bind on cell surface. They also analyze the effect of bafilomycinA1 (which blocks acidification of endosomes) and wiskostatin (inhibits actin polymerization dependent membrane remodeling) on enterovirus release into the cytoplasm. The manuscript is overall well written with clear figures. My specific comments are below:

1. The authors state that membrane warping and rupture of endosomes seen in infected cells is due to external pulling by cellular proteins. Authors also observe endosomal rupture in uninfected cells to support their claim. However, in line 246 of the manuscript, the authors also state 'All the studied enteroviruses induce endosome disruption', suggesting that the viruses or their proteins have a role in causing endosome disruption. It would be good to clarify this statement in the manuscript.
2. In line 106, the authors state "we show that a fraction of VLDL is released into the cytoplasm by endosome rupture via a mechanism morphologically similar to that utilized by enteroviruses. This provides evidence that VP4 and N-termini of VP1 of enteroviruses are not required for the disruption of endosomes, which releases enteroviruses into the cytoplasm". This sentence is not accurate. By studying the VLDL release into cytoplasm via endosome rupture, what the authors can conclude is that endosomal uptake and rupture is a fundamental cellular process which does not require VP4 and VP1 proteins. However, this does not 'provide evidence' that VP4 and VP1 have no role in endosomal rupture during enterovirus infection. Authors do not specifically label or track the position of the viral proteins to make a conclusive statement in this regard.
3. In the introduction, the authors mention that low pH and receptor binding as reasons for virion particle activation and genome release in enteroviruses. So, how do they justify their observation that only some particles release genome in the endosome at low pH conditions? From their observations, low pH and receptor binding alone does not trigger genome release in enteroviruses. In that case, can the authors briefly discuss possible reasons for this difference, as in what other factors may influence this process?
4. The authors postulate that virions releasing genome in cytoplasm are more likely to be infectious due to the lesser amount of RNases. But have the authors observed any genome release from enterovirus particles in the cytoplasm? Is it possible that the full-virion particles released into the cytoplasm without releasing genome are dead-end and cannot release genome anymore?
5. Do the authors observe empty endosomes (as in no virus inside) undergoing rupture in the virus-infected cells?
6. Is it possible that the virus/viral proteins may indirectly promote faster endosomal rupture on endocytosis?
7. The final ending paragraph of the manuscript needs to be re-written. Based on the authors' claims earlier in the manuscript, endosomal rupture is primarily a cellular function which is not influenced by the viral proteins. In that case, inhibiting a fundamental cellular process as a therapeutic intervention may not be a feasible strategy. Also, in this study,

there is no relationship explored for links between endosomal rupture and receptor protein endocytosed. Thus, suggesting to use nanoparticles that target enterovirus receptors is not a viable strategy but speculation. The manuscript is a step forward in figuring out the pathway for enterovirus cellular entry. It is a valuable study for its fundamental contributions and can be concluded as such.

Minor comments:

1. In Figure 3, what is indicated by cyan arrowheads? It is not mentioned in the legend. There is also a mistake in panel descriptions. Panel c is described as (b) in the legend, and panel d as (c).
2. In Figure 4 legend, panel e description says 'the odds of...', shouldn't it be 'instances of' or something similar?

Reviewer #2

(Remarks to the Author)

This well-written manuscript describes cryo-electron tomographic studies of enteroviruses infecting whole cells at various time points. The authors show clear images of enteroviruses within cells, both in endosomes and in the cytoplasm, and have clear visual evidence of actin-associated membrane deforming features that seem to rupture endosomes. The authors describe experiments they carried out to support these observations, and the theory that enteroviruses deliver their genomes into the cytoplasm through endosomal rupture.

I enjoyed reading this manuscript, and I think it absolutely should be published, but after some degree of carefully considered rephrasing. The model presented here is one possible mechanism of viral cell entry and infection, though some evidence in the literature points to other models as well. In general, I would welcome a more robust discussion of how this fits in to the larger picture, and what definitive experiments (which are likely outside the scope of this work) might answer these questions.

My particular comments are as follows:

In the Abstract: "Since the studied enteroviruses employ different receptors for cell entry but are all delivered into the cytoplasm by cell-mediated endosome disruption, it is likely that most if not all enteroviruses, and probably numerous other viruses from the family Picornaviridae, utilize endosome rupture to infect cells."

This statement is a bold claim, and constitutes the center of the questions I have about this manuscript. The authors seem to have demonstrated - quite effectively - that endosomal rupture does indeed happen (including with controls). In my view, they have not conclusively demonstrated that this is the primary mechanism of enteroviral infection in cells - though I will concede it is a possible mechanism. The authors state that an MOI of 29 was used, but a quick calculation based on the images casts doubt on this number. One tomogram shows over 200 virus particles, making the conservative estimate of the MOI on the cell to be 100, and possibly much higher.

- in line 722 the authors write: "Therefore, an average cell was infected by 29,000 virus particles. The infectious unit to particle ratio of enteroviruses is about 1 to 103 (66, 67). Therefore, the cells in our cryo-ET experiments were infected with a multiplicity of infection of 29."

More accurately, they should write, "the MOI was between 29 and 29000". And an MOI of 29000 is very high indeed! Even inert particles are internalized when enough are present in the medium. So how can we tell that this mechanism is physiologically relevant?

I sympathize with the desire to strike a balance between seeing events within the cell and finding conditions that reflect a physiological case. But in circumstances where a very high MOI is used, the authors are cautioned to temper their statements on definitive mechanism, given the wash of different endocytic compartments (coated, ruptured, unruptured) that are evident in the data.

Wiskostatin does not only inhibit membrane remodeling, it also inhibits endosomal trafficking and decreases cellular ATP levels, either of which could lead to endosomal conditions unsuitable for uncoating. Are the ionic conditions in the inhibited endosomes the same? This is important as these have been the classical triggers for uncoating of enteroviruses - the very model you are trying to unseat. I suggest a more thorough discussion

Line 94 - "seven to 60 minutes post infection" - this seems to be a useful range for studying

Line 106 - "However, after endocytosis, neither of the studied enteroviruses remained bound to the endosome membrane." How do you reconcile this with evidence to the contrary from Brandenburg et al? Later you (correctly) describe that the enteroviruses detach from the receptor. But letting go of the receptor and being separate from the membrane are distinct in a physiological context. The same membrane-binding effect of the VP1 peptide that was shown to disrupt membranes in high concentrations in Ref 44 is likely to anchor the capsid to the membrane. How is this explained?

Line 126 - "Since we did not observe the binding of the particles of any of the studied enteroviruses to endosome membranes (Fig. 2, S1-3), it is unlikely that they form an opening in a capsid connected to a transmembrane pore for genome delivery into the cytoplasm in vivo."

As a reviewer, it is very difficult to draw conclusions about the proximity of a membrane from a slice through a tomogram,

particularly when the top and bottom parts of the membrane are very poorly resolved by electron tomography. The packing of the particles suggests there is a degree of confinement - perhaps induced by the membrane. Indeed, Fig S2b & d show particles arrayed near a membrane. An assessment of the distance of the viruses to the nearest membrane feature would go a long way in supporting this statement.

Line 131 - Membrane warping. These images are spectacular! This is clear evidence of a membrane tensile force.

Line 149 - ruptured endosomes: Again, these images are simply wonderful. They clearly show ruptured membranes, and viruses leaking into the cell.

line 157 - "Therefore, it is likely that most if not all enteroviruses use endosome rupture to reach the cytoplasm." I would very strongly recommend modifying this to "... enteroviruses CAN use endosome rupture..."

line 165 - "It is possible that the virus particles and cell membranes were affected by the isolation procedure and may not represent the rhinovirus 2 infection process in vivo."

This is indeed a possibility, but an equal level of scrutiny must be applied to the experiments presented here. Such a high Multiplicity of Infection will see every conceivable mechanism of material internalisation from the cell surface, and membrane rupture may not represent the predominant, or even likely, physiological process in the context of an infection in humans. This MUST be highlighted in the manuscript for the sake of completeness.

line 207 - "This provides evidence that VP4 and N-termini of VP1 of enteroviruses are not required for the disruption of endosomes, which releases enteroviruses into the cytoplasm."

I agree that you seem to have found a mechanism for endosomal membrane disruption. And that your images of VLDL and enterovirus endosomal disruption appear to use the same mechanism. This by itself is a remarkable finding!

line 294 - The authors point out the independence of endosomal rupture on VP4 and the N-term of VP1, yet both of these are required for enteroviral infection. How do the authors reconcile this? Some attention should be given to this, or, at the very least, a discussion point should be how this can be (needs to be) reconciled in future studies.

Fig S4 - How many uncertain localizations where there?

Fig S8 - The curves are not labeled or described (Phase randomized... etc)

Reviewer #3

(Remarks to the Author)

The authors have addressed the thorny problem of viral genome release and the initiation of infection using cryo-Electron Tomography on cells at early stages post infection with a number of enteroviruses from the picornavirus family. In addition to infection of untreated cells, drugs that alter endosome related processes were also used in attempts to dissect the location and timing of the uncoating process. The results presented clearly show the presence of both native virus particles and empty particles within endosomal vesicles and in the cytoplasm. The fundamental question is which of these compartments is the site of functional genome release that initiates the infection cycle?

A basic problem that makes this such a difficult question to answer is that as the particle/plaque forming unit (PFU) ratio is notoriously high for these viruses, typically 100-1000/1. If 99-99.9% of infecting virus particles fail to initiate infection, how can the particles initiating productive infection be identified? In addition to the difficulties in identifying the process(es) which result in productive infection as a consequence of the high particle/PFU ratios inherent in these viruses, the authors had to infect cells with high multiplicities of infection (MOIs) in order to visualise sufficient numbers of particles for analysis.

The results presented beautifully illustrate the presence of both virus particles and empty particles within endosomes of infected cells and tomographic analysis failed to show direct interactions of particles with endosomal membranes. This contrasts with in vitro studies showing the direct binding of particles thought to be intermediates in the entry process with membranes. However, attempts to detect structurally such intermediate particles failed possibly due to their transient nature in vivo. There is clear evidence demonstrated in the micrographs that endosomes can disrupt to discharge their contents of virions and empty particles into the cytoplasm. Furthermore it was shown that such endosome disruption is not uniquely induced by virus infection as disrupted endosomes were observed in control cells and in cells treated with VLDL complexes. Wikostatin, an inhibitor of membrane remodelling, was shown to reduce infection by ECHO18 and increased the ratio of empty particles present in endosomes. This is taken as evidence against genome transmission across the membranes and into the cytoplasm. However, it also feasible that wikostatin might prevent such genome transfer by methods independent of endosome disruption such as preventing the establishment of transmembrane channels. It is interesting that this part of the study was conducted with ECHO18, which seemed to differ from the other viruses studied in the extent to which empty particles were seen in the cytoplasm. It is unclear on what basis viruses were selected for different aspects of the study. Another interesting theoretical consequence of the conclusion that virus uncoating occurs readily in the cytoplasm is the question of how newly assembled virions at the end of the growth cycle are protected from such uncoating.

This study is almost entirely based on structural analysis of virus and virus related particles during the infection process and it is a pity that some more biological/biochemical investigations could not be included. For example, assessment of genome integrity during the early stages of infection would have been a valuable addition to the study. It would have also been useful to discuss the more biochemical study of the poliovirus entry process conducted by Brandenburg, et al. It might also be interesting to examine the effects of uncoating inhibiting pocket factor analogue drugs on virus entry.

A number of quantitative aspects of the study should be clarified. These include:-

What were the numbers/proportions of endosomes displaying specific characteristics such as distortion and in the process of

disruption?

How were the particle numbers in the different virus samples assessed?

Were any attempts made to measure the particle/PFU ratios?

In conclusion this is an interesting addition to the debate on the mechanism of enterovirus cell entry and infection. It contrasts in some ways with other models of the process but it is entirely possible that alternative pathways may be followed depending on the specific virus studies and the precise conditions of the experimentation.

Version 1:

Reviewer comments:

Reviewer #1

(Remarks to the Author)

The authors have responded to my comments satisfactorily. I have no further revisions or comments.

Reviewer #2

(Remarks to the Author)

I am satisfied with the changes implemented. In my view, this manuscript should be published.

Reviewer #3

(Remarks to the Author)

In the revised version of their manuscript, Plevka, et al have adequately addressed the major concerns I and the others reviews had about the original version. The manuscript reports a beautifully conducted study of the intracellular location of enterovirus particles shortly after infection and the mechanisms involved to facilitate the infection process are implied from these morphological observations. The major criticism of the original version is that a number of important caveats to the proposed mechanism of infection via endosome disruption were not adequately discussed.

The revised version is modulated sufficiently to provide a more nuanced interpretation of their results and allows a more balanced evaluation of the contribution of their studies to the general discussion of how these viruses infect susceptible cells.

The reviewer's comments are in blue italics, and our responses are in bold black font. Please note that the line numbers in this document refer to the manuscript and supplementary material file with tracked changes, which was submitted as supplementary file for the revision process.

Reviewer #1 (Remarks to the Author):

The manuscript by Ishemgulova et.al., focuses on imaging entry by enteroviruses into cells. Direct data on how non-enveloped viruses, such as enteroviruses, deliver their genome into the cytoplasm in cells is sparsely available. In this work, the authors have used cryo-electron tomography and imaged thin, cellular extensions at the periphery of cos-7 cells to observe multiple endocytosed enteroviruses and their release into the cytoplasm. The authors have used rhinovirus 2, echovirus 18, echovirus 30, and enterovirus 71 viruses in this study and imaged the infected cell peripheries at increasing time points post-infection. Based on their results, the authors put forth a general mechanism for enterovirus entry into cellular cytoplasm and discuss that these viruses exploit a cellular mechanism for membrane remodeling that does not directly involve any of the viral proteins for endosomal release. Enterovirus particles and genome escape into the cytoplasm by endosome rupture, irrespective of which receptor they bind on cell surface. They also analyze the effect of bafilomycinA1 (which blocks acidification of endosomes) and wiskostatin (inhibits actin polymerization dependent membrane remodeling) on enterovirus release into the cytoplasm. The manuscript is overall well written with clear figures. My specific comments are below:

1. The authors state that membrane warping and rupture of endosomes seen in infected cells is due to external pulling by cellular proteins. Authors also observe endosomal rupture in uninfected cells to support their claim. However, in line 246 of the manuscript, the authors also state 'All the studied enteroviruses induce endosome disruption', suggesting that the viruses or their proteins have a role in causing endosome disruption. It would be good to clarify this statement in the manuscript.

A: Thank you. The statement was indeed misleading. We have now re-written the sentence (lines 373-374):

"Some enterovirus-containing endosomes ruptured (Fig. 1, S1-3, Movie S1), releasing virions and free genomes into the cytoplasm, which may cause infection."

2. In line 106, the authors state "we show that a fraction of VLDL is released into the cytoplasm by endosome rupture via a mechanism morphologically similar to that utilized by enteroviruses. This provides evidence that VP4 and N-termini of VP1 of enteroviruses are not required for the disruption of endosomes, which releases enteroviruses into the cytoplasm". This sentence is not accurate. By studying the VLDL release into cytoplasm via endosome rupture, what the authors can conclude is that endosomal uptake and rupture is a fundamental cellular process which does not require VP4 and VP1 proteins. However, this does not 'provide evidence' that VP4 and VP1 have no role in endosomal rupture during enterovirus infection. Authors do not specifically label or track the position of the viral proteins to make a conclusive statement in this regard.

A: Thank you, we have now corrected the interpretation of our results as indicated (lines 270-272):

"This provides evidence that endosomal rupture is a fundamental cellular process, which does not require enterovirus VP4 and VP1."

3. In the introduction, the authors mention that low pH and receptor binding as reasons for virion particle activation and genome release in enteroviruses. So, how do they justify their observation that only some particles release genome in the endosome at low pH conditions? From their observations, low pH and receptor binding alone does not trigger genome release in enteroviruses. In that case, can the authors briefly discuss possible reasons for this difference, as in what other factors may influence this process?

A: We have now included the requested discussion in the Introduction (lines 65-68):

“The reactions of virions of rhinovirus 14 to the genome-inducing environmental stimuli were shown to be heterogeneous – some of the virions remained in native form, while others converted to activated particles or released their genomes (33).”

And in the Results and Discussion (lines 350-364):

“The variability in the location and timing of genome release among individual enterovirus particles is probably caused by the variability in environmental stimuli experienced by individual particles and by internal differences between the particles. Individual particles bind to distinct numbers of receptors at the plasma membrane, and therefore receive distinct levels of stimulation (33, 81). Furthermore, particles in different endosomes are exposed to distinct pH levels and ionic compositions, depending on the maturation state of the particular endosome (82). The duration of their exposure to those endosome conditions also differs. Enterovirus virions may differ in the arrangement of packaged genomes and contain variable amounts of polyamines and inorganic cations that neutralize the repulsive interactions of virus RNA (83). Finally, the timing of genome release from a particular particle is subject to capsid breathing and genome rearrangements caused by thermal motions. It is possible that at least some of the genome-containing particles observed in our cryo-EM experiments in the cytoplasm would not be able to release their genomes and initiate infection. The inability of some of the particles to release their genomes may contribute to the high ratio of particles to infectious units of enteroviruses (33, 72, 73).”

4. The authors postulate that virions releasing genome in cytoplasm are more likely to be infectious due to the lesser amount of RNases. But have the authors observed any genome release from enterovirus particles in the cytoplasm?

A: We observed higher fraction of empty capsids among particles in the cytoplasm than among those in the endosomes, which provides indirect evidence that genome release occurs also in the cytoplasm. The evidence is summarized in Fig. S4 and discussed in the manuscript (lines 304-346):

“Here we show that both endosomes and the cytoplasm of infected cos-7 cells contained full and empty particles of the studied enteroviruses (Fig. 1, 2, S1-4, Movie S1). In most cases, the cytoplasm contained a higher fraction of empty particles than endosomes (Fig. S4), which provides evidence that enteroviruses release their genomes in the endosomes and cytoplasm, as indicated previously (15).”

Is it possible that the full-virion particles released into the cytoplasm without releasing genome are dead-end and cannot release genome anymore?

A: Yes, it is likely that at least some of the genome-containing particles observed in our cryo-EM experiments may not release their genomes and thus become dead end products. We have now extended discussion of these aspects (lines 361-364):

“It is possible that at least some of the genome-containing particles observed in our cryo-EM experiments in the cytoplasm would not be able to release their genomes and initiate infection. The inability of some of the particles to release their genomes may contribute to the high ratio of particles to infectious units of enteroviruses (33, 72, 73).”

5. Do the authors observe empty endosomes (as in no virus inside) undergoing rupture in the virus-infected cells?

A: Yes, we observed empty endosomes undergoing rupture in infected cells. We have now included a new Fig. S8 showing such endosomes and commented on this observation in the text (lines 245-247):

“Furthermore, non-infected control cells contain a fraction of endosomes with ruptured membranes (Fig. 3b, S7) and infected cells contain ruptured endosomes with no virus inside (Fig. S8, Table S1).”

Please note that upon request from reviewer #3 we have now included Table S1 summarizing the number of occurrences of endosomes with smooth, warped, and ruptured membranes in control and infected cells at various times post infection. This also includes the evaluation of the state of endosomes that did not contain virus in infected cells.

6. Is it possible that the virus/viral proteins may indirectly promote faster endosomal rupture on endocytosis?

A: The interaction of enterovirus capsids with receptors may induce receptor aggregation and activation and co-endocytosis of the receptors with virus particles. We speculate that signaling from the receptors embedded in membranes of virus-containing endosomes may promote endosome rupture. We have now included additional discussion of this topic (lines 247-251):

“This indicates that endosome disruption can be triggered by a cellular mechanism without the direct involvement of virus components. Enterovirus cell entry probably induces the concentration of virus-specific receptors in endosomes, which may stimulate signaling that activates cellular pathways that mediate endosomal rupture.”

7. The final ending paragraph of the manuscript needs to be re-written. Based on the authors' claims earlier in the manuscript, endosomal rupture is primarily a cellular function which is not influenced by the viral proteins. In that case, inhibiting a fundamental cellular process as a therapeutic intervention may not be a feasible strategy. Also, in this study, there is no relationship explored for links between endosomal rupture and receptor protein endocytosed. Thus, suggesting to use nanoparticles that target enterovirus receptors is not a viable strategy but speculation. The manuscript is a step forward in figuring out the pathway for enterovirus cellular entry. It is a valuable study for its fundamental contributions and can be concluded as such.

A: We have now removed the last paragraph.

Minor comments:

1. In Figure 3, what is indicated by cyan arrowheads? It is not mentioned in the legend. There is also a mistake in panel descriptions. Panel c is described as (b) in the legend, and panel d as (c).

A: Thank you. Cyan arrowheads indicate actin microfilaments. We have now corrected the figure legend (Line 541):

“Actin microfilaments are indicated by cyan arrowheads.”

2. In Figure 4 legend, panel e description says ‘the odds of...’, shouldn’t it be ‘instances of’ or something similar?

A: Thank you, we have now changed “the odds of” to “the probability of” (lines 561-563):

(e) Plot comparing differences in the probability of genome release from enterovirus particles in endosomes and cytoplasm of wiskostatin and bafilomycin A1-treated cells relative to the infection of untreated cells.”

Reviewer #2 (Remarks to the Author):

This well-written manuscript describes cryo-electron tomographic studies of enteroviruses infecting whole cells at various time points. The authors show clear images of enteroviruses within cells, both in endosomes and in the cytoplasm, and have clear visual evidence of actin-associated membrane deforming features that seem to rupture endosomes. The authors describe experiments they carried out to support these observations, and the theory that enteroviruses deliver their genomes into the cytoplasm through endosomal rupture.

I enjoyed reading this manuscript, and I think it absolutely should be published, but after some degree of carefully considered rephrasing. The model presented here is one possible mechanism of viral cell entry and infection, though some evidence in the literature points to other models as well. In general, I would welcome a more robust discussion of how this fits in to the larger picture, and what definitive experiments (which are likely outside the scope of this work) might answer these questions.

My particular comments are as follows:

In the Abstract: "Since the studied enteroviruses employ different receptors for cell entry but are all delivered into the cytoplasm by cell-mediated endosome disruption, it is likely that most if not all enteroviruses, and probably numerous other viruses from the family Picornaviridae, utilize endosome rupture to infect cells."

This statement is a bold claim, and constitutes the center of the questions I have about this manuscript. The authors seem to have demonstrated - quite effectively - that endosomal rupture does indeed happen (including with controls). In my view, they have not conclusively demonstrated that this is the primary mechanism of enteroviral infection in cells - though I will concede it is a possible mechanism. The authors state that an MOI of 29 was used, but a quick calculation based on the images casts doubt on this number. One tomogram shows over 200 virus particles, making the conservative estimate of the MOI on the cell to be 100, and possibly much higher.

A: We address this comment in two parts:

- 1.) We have now modified the sentence so that it does not exclude the possibility of enteroviruses utilizing other mechanisms to enter cells. This is the same modification as suggested by reviewer #2 for a similar statement in the Results and discussion section (lines 35-38):**
"Since the studied enteroviruses employ different receptors for cell entry but are all delivered into the cytoplasm by cell-mediated endosome disruption, it is likely that most if not all enteroviruses, and probably numerous other viruses from the family Picornaviridae, can utilize endosome rupture to infect cells."
- 2.) Please note that the tomogram sections included in the figures in the manuscript were selected to show virus particles, as they are the objects of our interest. Thus, we do not show sections of tomograms that were devoid or nearly devoid of virus particles. We used multiple whole tomograms to determine virus concentration in infected cells. Please see Table S2 for details.**

- in line 722 the authors write: "Therefore, an average cell was infected by 29,000 virus particles. The infectious unit to particle ratio of enteroviruses is about 1 to 103 (66, 67).

Therefore, the cells in our cryo-ET experiments were infected with a multiplicity of infection of 29.” More accurately, they should write, “the MOI was between 29 and 29000”. And an MOI of 29000 is very high indeed! Even inert particles are internalized when enough are present in the medium. So how can we tell that this mechanism is physiologically relevant?

A: Published evidence indicates that enteroviruses have PFU to particle ratio of about 1 : 1,000.

C. E. Schwerdt, J. Fogh, The ratio of physical particles per infectious unit observed for poliomyelitis viruses. Virology 4, 41-52 (1957).

S. J. Flint, Principles of virology : molecular biology, pathogenesis, and control of animal viruses. (ASM Press, Washington, D.C., ed. 2nd, 2004), pp. xxvi, 918 p.

C. R. Buttner, R. Spurny, T. Fuzik, P. Plevka, Cryo-electron microscopy and image classification reveal the existence and structure of the coxsackievirus A6 virion. Commun Biol 5, 898 (2022).

Therefore, stating that the MOI used was between 29 and 29,000 would be misleading.

We have now re-written the text in materials and methods section to clearly state the PFU to particle ratio of enteroviruses (lines 934-935):

“For enteroviruses, the ratio of infectious units to particles is approximately 1 : 1000 (72-74).”

We agree that careful interpretation of results is prudent and essential. – Please see our response to the next point.

I sympathize with the desire to strike a balance between seeing events within the cell and finding conditions that reflect a physiological case. But in circumstances where a very high MOI is used, the authors are cautioned to temper their statements on definitive mechanism, given the wash of different endocytic compartments (coated, ruptured, unruptured) that are evident in the data.

A: We have now tempered our statements regarding the mechanisms employed by enteroviruses to enter cells:

Lines 453-457:

“Since the studied enteroviruses employ distinct receptors for cell entry but are all delivered into the cytoplasm by cell-mediated endosome disruption, we propose that most if not all enteroviruses, and probably numerous other viruses from the family Picornaviridae, can use endosome rupture to reach the cytoplasm.”

Lines 223-231:

“To enable the visualization of enterovirus infection by cryo-ET, the cells were infected using a multiplicity of infection of 29 (please see Materials and methods for details). This high virus dose was required to enable the observation of multiple enterovirus particles in high-magnification tomograms, each of which captured only 0.17% of a cell. Therefore, our cryo-ET analyses do not provide proof that endosome disruption is the only enterovirus infection pathway. The high multiplicity of infection probably enabled enteroviruses to enter cells using multiple mechanisms of internalization, and the observed membrane rupture may not represent the most abundant physiological process in the context of a low multiplicity of infection.”

Lines 344-349:

“In most cases, the cytoplasm contained a higher fraction of empty particles than endosomes (Fig. S4), which provides evidence that enteroviruses release their genomes in the endosomes and cytoplasm, as indicated previously (15). However, there is evidence that the productive genome release of poliovirus occurs across the membrane of vesicles or tightly sealed membrane invaginations (57). It is possible that enteroviruses employ multiple pathways for genome delivery, as discussed below.”

Lines 447-450:

“N-WASP-mediated actin polymerization (67, 68) contributes to the warping and rupture of enterovirus-containing endosomes, demonstrating that enteroviruses can utilize a cellular mechanism for gaining entry into the cytoplasm (Fig. 5).”

Lines 35-38:

“Since the studied enteroviruses employ different receptors for cell entry but are all delivered into the cytoplasm by cell-mediated endosome disruption, it is likely that most if not all enteroviruses, and probably numerous other viruses from the family Picornaviridae, can utilize endosome rupture to infect cells.”

Wiskostatin does not only inhibit membrane remodeling, it also inhibits endosomal trafficking and decreases cellular ATP levels, either of which could lead to endosomal conditions unsuitable for uncoating. Are the ionic conditions in the inhibited endosomes the same? This is important as these have been the classical triggers for uncoating of enteroviruses - the very model you are trying to unseat. I suggest a more thorough discussion.

A: Our results are consistent with previous observations that receptor binding, acidic pH, and changes in ion composition induce genome release of enteroviruses. To present our results in context of the previous observations indicated by reviewer #2 we have now extended the description of the effects of wiskostatin on cells (lines 277-296):

“The binding of wiskostatin to N-WASP stabilizes the protein in autoinhibited conformation, prevents it from activating the Arp2/3 complex, and thus blocks processes of cellular membrane remodeling that depend on actin polymerization (77, 78). However, wiskostatin treatment also causes a decrease in cellular ATP levels and inhibits other, N-WASP-independent functions (79). Additionally to HRV2, echovirus 18 was selected for the wiskostatin inhibition studies because it propagated in cos7 cells most efficiently of the analyzed viruses. In wiskostatin-treated cells most of the particles were trapped inside oblique endosomes; however, some were released into the cytoplasm (Fig. 4). The infection of cells pre-treated with wiskostatin increased the odds of genome release inside endosomes 10-fold relative to untreated cells (Fig. 4e, Table S2). The inhibition of endosomal membrane remodeling by wiskostatin probably resulted in a prolonged retention of enterovirus particles in endosomes, where they released genomes. Relative to one-hour post-infection, echovirus 18 replicated its genome 20- and 35-fold, seven and nine hours post-infection, respectively (Fig. S9). In contrast, in cos-7 cells pretreated with wiskostatin, the relative replication of echovirus 18 did not exceed 5-fold both seven and nine hours post-infection (Fig. S9). We speculate that the reduced efficiency of genome replication was caused by predominant genome release in endosomes, where the virus genomes may have been degraded by RNases. However, the efficiency of echovirus 18 genome replication may have been also reduced by decreased levels of proteosynthesis

and RNA synthesis due to the low cellular ATP levels, which are a side effect of wiskostatin treatment (79).”

Line 94 - “seven to 60 minutes post infection” - this seems to be a useful range for studying

A: Thank you.

Line 106 - “However, after endocytosis, neither of the studied enteroviruses remained bound to the endosome membrane.” How do you reconcile this with evidence to the contrary from Brandenburg et al? Later you (correctly) describe that the enteroviruses detach from the receptor. But letting go of the receptor and being separate from the membrane are distinct in a physiological context. The same membrane-binding effect of the VP1 peptide that was shown to disrupt membranes in high concentrations in Ref 44 is likely to anchor the capsid to the membrane. How is this explained?

A: We have now changed the sentence to state that the studied enteroviruses detached from receptors (lines 110-111):

“However, after endocytosis, all of the studied enteroviruses detached from receptors (Fig. 2a-f, S5, Movie S2).”

We have now included additional discussion to reconcile our observations with the results and hypotheses presented by: Brandenburg B, Lee LY, Lakadamyali M, Rust MJ, Zhuang X, Hogle JM. Imaging poliovirus entry in live cells. PLoS Biol. 2007 Jul;5(7):e183. and references in this article (lines 121-144):

“It has been shown previously that the N-termini of VP1 subunits, which are exposed at the surface of activated particles, can mediate membrane binding and could even induce membrane disruption (27, 29, 52-57). Furthermore, VP4 subunits released from enterovirus activated particles were shown to form pores in membranes (58). In vitro studies of poliovirus and HRV2 have shown the binding of activated particles to liposomes, and it was speculated that the transfer of RNA from the activated particles is mediated by umbilical connections up to 5 nm long (37, 41, 45, 59). However, cryo-tomograms of infected cells show nearly all enterovirus particles located more than 5 nm from the closest endosome membrane (Fig. 2, S1-3, S6). With the exception of tight packing of particles in the minority of endosomes, enterovirus particles were distributed randomly in endosomes, and their mean distance from the closest membrane was 36 nm +/- 23 nm (Fig. S6). These contrasting results indicate that enteroviruses may employ not only various receptors to attach to cells and different endocytic pathways to gain entry into host cells, but also multiple mechanisms to breach endosome membranes (15, 57).”

Line 126 - “Since we did not observe the binding of the particles of any of the studied enteroviruses to endosome membranes (Fig. 2, S1-3), it is unlikely that they form an opening in a capsid connected to a transmembrane pore for genome delivery into the cytoplasm in vivo.”

As a reviewer, it is very difficult to draw conclusions about the proximity of a membrane from a slice through a tomogram, particularly when the top and bottom parts of the membrane are very poorly resolved by electron tomography. The packing of the particles suggests there is a degree of confinement - perhaps induced by the membrane. Indeed, Fig S2b & d show particles arrayed near a membrane. An assessment of the distance of the viruses to the nearest membrane feature would go a long way in supporting this statement.

A: We have now re-written the corresponding part of the text and included new Fig. S6 analyzing distances of enterovirus particles from endosome membranes (lines 127-131): “However, cryo-tomograms of infected cells show nearly all enterovirus particles located more than 5 nm from the closest endosome membrane (Fig. 2, S1-3, S6). With the exception of tight packing of particles in the minority of endosomes, enterovirus particles were distributed randomly in endosomes, and their mean distance from the closest membrane was 36 nm +/- 23 nm (Fig. S6).”

Line 131 - Membrane warping. These images are spectacular! This is clear evidence of a membrane tensile force.

A: Thank you.

Line 149 - ruptured endosomes: Again, these images are simply wonderful. They clearly show ruptured membranes, and viruses leaking into the cell.

A: Thank you.

line 157 - “Therefore, it is likely that most if not all enteroviruses use endosome rupture to reach the cytoplasm.”

I would very strongly recommend modifying this to “... enteroviruses CAN use endosome rupture...”

A: We have now modified the sentence according to reviewer’s suggestion (lines 172-173): “Therefore, it is likely that most, if not all enteroviruses can use endosome rupture to reach the cytoplasm.”

line 165 - “It is possible that the virus particles and cell membranes were affected by the isolation procedure and may not represent the rhinovirus 2 infection process in vivo.” This is indeed a possibility, but an equal level of scrutiny must be applied to the experiments presented here. Such a high Multiplicity of Infection will see every conceivable mechanism of material internalisation from the cell surface, and membrane rupture may not represent the predominant, or even likely, physiological process in the context of an infection in humans. This MUST be highlighted in the manuscript for the sake of completeness.

A: We have now included a conditioning statement in the manuscript (lines 228-231): “The high multiplicity of infection probably enabled enteroviruses to enter cells using multiple mechanisms of internalization, and the observed membrane rupture may not represent the most abundant physiological process in the context of a low multiplicity of infection.”

line 207 - “This provides evidence that VP4 and N-termini of VP1 of enteroviruses are not required for the disruption of endosomes, which releases enteroviruses into the cytoplasm.” I agree that you seem to have found a mechanism for endosomal membrane disruption. And that your images of VLDL and enterovirus endosomal disruption appear to use the same mechanism. This by itself is a remarkable finding!

A: Thank you.

line 294 - The authors point out the independence of endosomal rupture on VP4 and the N-term of VP1, yet both of these are required for enteroviral infection. How do the authors

reconcile this? Some attention should be given to this, or, at the very least, a discussion point should be how this can be (needs to be) reconciled in future studies.

A: We have now re-written this section to avoid this contentious point (lines 447-450):

“N-WASP-mediated actin polymerization (67, 68) contributes to the warping and rupture of enterovirus-containing endosomes, demonstrating that enteroviruses can utilize a cellular mechanism for gaining entry into the cytoplasm (Fig. 5).”

We have included extended discussion of the role of VP4 and N-terminus of VP1 in cell entry in response to the previous comments made by reviewer #2 (lines 121-144):

“It has been shown previously that the N-termini of VP1 subunits, which are exposed at the surface of activated particles, can mediate membrane binding and could even induce membrane disruption (27, 29, 52-57). Furthermore, VP4 subunits released from enterovirus activated particles were shown to form pores in membranes (58). In vitro studies of poliovirus and HRV2 have shown the binding of activated particles to liposomes, and it was speculated that the transfer of RNA from the activated particles is mediated by umbilical connections up to 5 nm long (37, 41, 45, 59). However, cryo-tomograms of infected cells show nearly all enterovirus particles located more than 5 nm from the closest endosome membrane (Fig. 2, S1-3, S6). With the exception of tight packing of particles in the minority of endosomes, enterovirus particles were distributed randomly in endosomes, and their mean distance from the closest membrane was 36 nm +/- 23 nm (Fig. S6). These contrasting results indicate that enteroviruses may employ not only various receptors to attach to cells and different endocytic pathways to gain entry into host cells, but also multiple mechanisms to breach endosome membranes (15, 57).”

Fig S4 - How many uncertain localizations where there?

A: 10% of particles had uncertain localization. We have now included this information in the figure legend (lines 1164-1165):

“Particles with uncertain localizations (10%) were omitted from the analyses.”

Fig S8 - The curves are not labeled or described (Phase randomized... etc)

A: Thank you, we have now included description of the curve colors in the figure legend (lines 1218-1220):

“Fourier shell correlation curves of Fourier shell correlation corrected half-maps (black), unmasked half-maps (green), masked half-maps (blue), and phase-randomized-masked half-maps (red) of individual cryo-EM reconstructions.”

Reviewer #3 (Remarks to the Author):

The authors have addressed the thorny problem of viral genome release and the initiation of infection using cryo-Electron Tomography on cells at early stages post infection with a number of enteroviruses from the picornavirus family. In addition to infection of untreated cells, drugs that alter endosome related processes were also used in attempts to dissect the location and timing of the uncoating process. The results presented clearly show the presence of both native virus particles and empty particles within endosomal vesicles and in the cytoplasm. The fundamental question is which of these compartments is the site of functional genome release that initiates the infection cycle?

A: Our results do not allow us to discriminate which of the compartments is the site of genome release that leads to the productive infection. Previous results by Brandenburg et al. PLOS Biology 2007, indicate that productive genome release was from vesicles or tightly sealed membrane invaginations. However, Schober et al. JVI 1998 summarize the evidence that major and minor group rhinoviruses release their genomes at the different compartments. We have now expanded the discussion and comparison of our and previous results (lines 304-364):

“Here we show that both endosomes and the cytoplasm of infected cos-7 cells contained full and empty particles of the studied enteroviruses (Fig. 1, 2, S1-4, Movie S1). In most cases, the cytoplasm contained a higher fraction of empty particles than endosomes (Fig. S4), which provides evidence that enteroviruses release their genomes in the endosomes and cytoplasm, as indicated previously (15). However, there is evidence that the productive genome release of poliovirus occurs across the membrane of vesicles or tightly sealed membrane invaginations (57). It is possible that enteroviruses employ multiple pathways for genome delivery, as discussed below.

The variability in the location and timing of genome release among individual enterovirus particles is probably caused by the variability in environmental stimuli experienced by individual particles and by internal differences between the particles. Individual particles bind to distinct numbers of receptors at the plasma membrane, and therefore receive distinct levels of stimulation (33, 81). Furthermore, particles in different endosomes are exposed to distinct pH levels and ionic compositions, depending on the maturation state of the particular endosome (82). The duration of their exposure to those endosome conditions also differs. Enterovirus virions may differ in the arrangement of packaged genomes and contain variable amounts of polyamines and inorganic cations that neutralize the repulsive interactions of virus RNA (83). Finally, the timing of genome release from a particular particle is subject to capsid breathing and genome rearrangements caused by thermal motions. It is possible that at least some of the genome-containing particles observed in our cryo-EM experiments in the cytoplasm would not be able to release their genomes and initiate infection. The inability of some of the particles to release their genomes may contribute to the high ratio of particles to infectious units of enteroviruses (33, 72, 73).”

A basic problem that makes this such a difficult question to answer is that as the particle/plaque forming unit (PFU) ratio is notoriously high for these viruses, typically 100-1000/1. If 99-99.9% of infecting virus particles fail to initiate infection, how can the particles initiating productive infection be identified?

A: Indeed, we cannot determine which particles would cause infection. This is stated in the manuscript (lines 219-223):

“Cryo-ET provides snapshots of cell and virus structures; however, it does not allow monitoring of the progress of infection of individual virus particles over time. Therefore, it cannot be determined which, if any, of the observed particles would successfully initiate infection. Moreover, the infectious unit-to-particle ratio of most enteroviruses is about 1 to 1,000 (72-74).”

In addition to the difficulties in identifying the process(es) which result in productive infection as a consequence of the high particle/PFU ratios inherent in these viruses, the authors had to infect cells with high multiplicities of infection (MOIs) in order to visualise sufficient numbers of particles for analysis.

The results presented beautifully illustrate the presence of both virus particles and empty particles within endosomes of infected cells and tomographic analysis failed to show direct interactions of particles with endosomal membranes. This contrasts with in vitro studies showing the direct binding of particles thought to be intermediates in the entry process with membranes. However, attempts to detect structurally such intermediate particles failed possibly due to their transient nature in vivo. There is clear evidence demonstrated in the micrographs that endosomes can disrupt to discharge their contents of virions and empty particles into the cytoplasm. Furthermore it was shown that such endosome disruption is not uniquely induced by virus infection as disrupted endosomes were observed in control cells and in cells treated with VLDL complexes. Wiskostatin, an inhibitor of membrane remodelling, was shown to reduce infection by ECHO18 and increased the ratio of empty particles present in endosomes. This is taken as evidence against genome transmission across the membranes and into the cytoplasm. However, it also feasible that wiskostatin might prevent such genome transfer by methods independent of endosome disruption such as preventing the establishment of transmembrane channels.

A: Wiskostatin has multiple effects on cells, including reduction of cellular ATP levels, which may interfere with progress of enterovirus infection by other mechanisms that blocking the genome from reaching cytoplasm. We have now included discussion of these effects on enterovirus infection in the manuscript (lines 275-298):

“To test the hypothesis that endosome disruption, mediated by a cellular pathway, enables enteroviruses to reach the cytoplasm, we blocked post-endocytic membrane remodeling with wiskostatin. The binding of wiskostatin to N-WASP stabilizes the protein in autoinhibited conformation, prevents it from activating the Arp2/3 complex, and thus blocks processes of cellular membrane remodeling that depend on actin polymerization (77, 78). However, wiskostatin treatment also causes a decrease in cellular ATP levels and inhibits other, N-WASP-independent functions (79). Additionally to HRV2, echovirus 18 was selected for the wiskostatin inhibition studies because it propagated in cos7 cells most efficiently of the analyzed viruses. In wiskostatin-treated cells most of the particles were trapped inside oblique endosomes; however, some were released into the cytoplasm (Fig. 4). The infection of cells pre-treated with wiskostatin increased the odds of genome release inside endosomes 10-fold relative to untreated cells (Fig. 4e, Table S2). The inhibition of endosomal membrane remodeling by wiskostatin probably resulted in a prolonged retention of enterovirus particles in endosomes, where they released genomes. Relative to one-hour post-infection, echovirus 18 replicated its genome 20- and 35-fold, seven and nine hours post-infection, respectively (Fig. S9). In contrast, in cos-7 cells

pretreated with wiskostatin, the relative replication of echovirus 18 did not exceed 5-fold both seven and nine hours post-infection (Fig. S9). We speculate that the reduced efficiency of genome replication was caused by predominant genome release in endosomes, where the virus genomes may have been degraded by RNases. However, the efficiency of echovirus 18 genome replication may have been also reduced by decreased levels of proteosynthesis and RNA synthesis due to the low cellular ATP levels, which are a side effect of wiskostatin treatment (79). Nevertheless, our results demonstrate that N-WASP-mediated actin polymerization is important for the disruption of endosomes, which enables the delivery of enteroviruses into the cytoplasm.”

It is interesting that this part of the study was conducted with ECHO18, which seemed to differ from the other viruses studied in the extent to which empty particles were seen in the cytoplasm. It is unclear on what basis viruses were selected for different aspects of the study.

A: We selected echovirus 18 for the wiskostatin inhibition experiment because it propagated most efficiently of the studied viruses in cos7 cells. We have now included the explanation in the manuscript (lines 281-283):

“echovirus 18 was selected for the wiskostatin inhibition studies because it propagated in cos7 cells most efficiently of the analyzed viruses.”

Another interesting theoretical consequence of the conclusion that virus uncoating occurs readily in the cytoplasm is the question of how newly assembled virions at the end of the growth cycle are protected from such uncoating.

A: We have now included discussion of this aspect (lines 380-384):

“The possibility of the initiation of the infection by genome release in the cytoplasm raises the question of how newly assembled enterovirus virions are protected from premature uncoating. We speculate that the newly assembled virions do not release their genomes because they have not been stimulated by environmental factors including receptor binding, acidic pH, and an ion composition distinct from that of the cytoplasm (26-30).”

This study is almost entirely based on structural analysis of virus and virus related particles during the infection process and it is a pity that some more biological/biochemical investigations could not be included. For example, assessment of genome integrity during the early stages of infection would have been a valuable addition to the study. It would have also been useful to discuss the more biochemical study of the poliovirus entry process conducted by Brandenburg, et al. It might also be interesting to examine the effects of uncoating inhibiting pocket factor analogue drugs on virus entry.

A: We have now included extended discussion of Brandenburg et al. 2007 results and of the effects of capsid binding inhibitors (lines 121-144):

“It has been shown previously that the N-termini of VP1 subunits, which are exposed at the surface of activated particles, can mediate membrane binding and could even induce membrane disruption (27, 29, 52-57). Furthermore, VP4 subunits released from enterovirus activated particles were shown to form pores in membranes (58). In vitro studies of poliovirus and HRV2 have shown the binding of activated particles to liposomes, and it was speculated that the transfer of RNA from the activated particles is mediated by umbilical connections up to 5 nm long (37, 41, 45, 59). However, cryo-tomograms of infected cells show nearly all enterovirus particles located more than 5 nm from the closest endosome membrane (Fig. 2, S1-3, S6). With the exception of tight packing of

particles in the minority of endosomes, enterovirus particles were distributed randomly in endosomes, and their mean distance from the closest membrane was 36 nm +/- 23 nm (Fig. S6). These contrasting results indicate that enteroviruses may employ not only various receptors to attach to cells and different endocytic pathways to gain entry into host cells, but also multiple mechanisms to breach endosome membranes (15, 57).”

And (lines 346-364):

“However, there is evidence that the productive genome release of poliovirus occurs across the membrane of vesicles or tightly sealed membrane invaginations (57). It is possible that enteroviruses employ multiple pathways for genome delivery, as discussed below.

The variability in the location and timing of genome release among individual enterovirus particles is probably caused by the variability in environmental stimuli experienced by individual particles and by internal differences between the particles. Individual particles bind to distinct numbers of receptors at the plasma membrane, and therefore receive distinct levels of stimulation (33, 81). Furthermore, particles in different endosomes are exposed to distinct pH levels and ionic compositions, depending on the maturation state of the particular endosome (82). The duration of their exposure to those endosome conditions also differs. Enterovirus virions may differ in the arrangement of packaged genomes and contain variable amounts of polyamines and inorganic cations that neutralize the repulsive interactions of virus RNA (83). Finally, the timing of genome release from a particular particle is subject to capsid breathing and genome rearrangements caused by thermal motions. It is possible that at least some of the genome-containing particles observed in our cryo-EM experiments in the cytoplasm would not be able to release their genomes and initiate infection. The inability of some of the particles to release their genomes may contribute to the high ratio of particles to infectious units of enteroviruses (33, 72, 73).”

We have now included discussion of the effects of capsid-binding inhibitors on enterovirus entry (lines 394-400):

“Compounds binding to hydrophobic pockets of VP1 block the activation of enterovirus particles and genome release, and are efficient inhibitors of enterovirus infection (86-93). Some of these inhibitors alter the conformation of the base of the receptor binding “canyon,” forming the roof of the pocket, and block receptor binding (94). Other capsid binding inhibitors allow receptor binding but prevent the conversion of virions to activated particles (95-97). Virions that cannot release their genomes are probably degraded by the cells that endocytosed them.”

A number of quantitative aspects of the study should be clarified. These include: What were the numbers/proportions of endosomes displaying specific characteristics such as distortion and in the process of disruption?

A: We have now included a new Table S1 summarizing the numbers of observations of smooth, distorted, and ruptured endosomes in infected and control cells.

How were the particle numbers in the different virus samples assessed?

A: The numbers of particles in different samples analyzed by cryo-ET were determined by visual inspection and counting. We have now included this information in the manuscript (lines 1163-1164):

“The localizations of virus particles were determined by the visual inspection of electron tomograms of infected cells.”

Were any attempts made to measure the particle/PFU ratios?

A: We did not determine the PFU to particle ratios for the studied viruses.

In conclusion this is an interesting addition to the debate on the mechanism of enterovirus cell entry and infection. It contrasts in some ways with other models of the process but it is entirely possible that alternative pathways may be followed depending on the specific virus studies and the precise conditions of the experimentation.

A: Thank you.